# Probabilistic precipitation downscaling for ungauged mountain sites: a pilot study for the Hindu Kush Himalaya

Marc Girona-Mata[1,2], Andrew Orr[2], Martin Widmann[3], Daniel Bannister[4], Ghulam Hussain Dars[5], Scott Hosking[2,6], Jesse Norris[7], David Ocio[8], Tony Phillips[2], Jakob Steiner[9,10], and Richard E. Turner[1]

[1]Department of Engineering, University of Cambridge, Cambridge, UK
[2]British Antarctic Survey, UK Research and Innovation, Cambridge, UK
[3]School of Geography, Earth and Environmental Sciences, University of Birmingham, Birmingham, UK
[4]WTW Research Network, WTW, London, UK
[5]U.S.-Pakistan Center for Advanced Studies in Water, Mehran University of Engineering and Technology, Jamshoro, Pakistan
[6]The Alan Turing Institute, London, UK
[7]Atmospheric and Oceanic Sciences, University of California Los Angeles, Los Angeles, CA, USA
[8]Mott MacDonald, Cambridge, UK
[9]Institute of Geography and Regional Science, University of Graz, Graz, Austria
[10]Himalayan University Consortium, Lalitpur, Nepal

**Correspondence:** Marc Girona-Mata (mg963@cam.ac.uk)

**Abstract.** This study introduces a novel approach to post-processing (i.e., downscaling and bias-correcting) reanalysis-driven regional climate model daily precipitation that is capable of generalising to ungauged mountain locations by leveraging sparse in situ observations and a probabilistic regression framework. We call this post-processing approach Generalised Probabilistic Regression (GPR), and implement it using both generalised linear models and artificial neural networks (i.e., multilayer perceptrons). By testing the GPR post-processing approach across three Hindu Kush Himalaya basins with varying hydro-meteorological characteristics and four experiments, which are representative of real-world scenarios, we find it performs consistently much better than both raw regional climate model output and deterministic bias correction methods for generalising daily precipitation post-processing to ungauged locations. We also find that GPR models are flexible and can be trained using data from a single region or multiple regions combined together, without major impacts on model performance. Additionally, we show that the GPR approach results in superior skill for post-processing entirely ungauged regions, by leveraging data from other regions, as well as ungauged high-elevation ranges. This suggests that GPR models have potential for extending post-processing of daily precipitation to ungauged areas of HKH. Whilst multilayer perceptrons yield marginally improved results overall, generalised linear models are a robust choice particularly for data-scarce scenarios, i.e., post-processing extreme precipitation events and generalising to completely ungauged regions.

## 1 Introduction

The mountain ranges of High Mountain Asia, often referred to as the Water Towers of Asia (Immerzeel et al., 2010), are the source of many major rivers in South Asia, such as the Indus or the Ganges, supplying water resources to a rich diversity of terrestrial and marine ecosystems (Xu et al., 2019) and to approximately 2 billion people living in or directly downstream

of them (Bolch et al., 2012; Mukherji et al., 2019; Wester et al., 2019; Widmann et al., 2019). These resources are heavily reliant on precipitation caused primarily by large-scale atmospheric circulations, such as the Indian summer monsoon and winter westerly disturbances, interacting with the steep orography that characterises the southern rim of High Mountain Asia, comprising the Hindu Kush Himalaya (HKH) mountain ranges (Bookhagen and Burbank, 2010; Palazzi et al., 2013; Baudouin et al., 2020; Dimri et al., 2015). Yet, despite the large human populations depending on these resources for power, industry, tourism, farming, and domestic consumption, the contributions of rain and snow (and its ensuing melt) to these river systems are still poorly studied and little understood. This precipitation knowledge gap in HKH severely affects our ability to quantify its present-day water resources and associated stream flows (Immerzeel et al., 2015; Arfan et al., 2019; Li et al., 2018; Salzmann et al., 2014). Consequently, it constitutes the largest source of uncertainty when it comes to making effective and robust water management decisions (e.g., water infrastructure construction, water demand management) and their critical role in regulating regional water supply, as well as planning for the hydrological impacts of climate change (Chinnasamy et al., 2015; Momblanch et al., 2019; Wester et al., 2019; Nie et al., 2021; Orr et al., 2022).

Improving our understanding of precipitation in the HKH region is highly challenging (e.g., Wester et al., 2019; Sabin et al., 2020; Orr et al., 2022). In particular, the extreme orography that characterises this region results in large precipitation variations over small spatio-temporal scales, which are poorly understood due to the sparse and uneven rain and snow gauge network across the region (Archer and Fowler, 2004; Bannister et al., 2019; Immerzeel et al., 2015; Bookhagen and Burbank, 2010; Baudouin et al., 2020; Pritchard, 2021). For example, an area of around 566,000 km$^2$ above 4000 m elevation in HKH is currently represented by a single long-running gauge station in the Global Historical Climatology Network database (Pritchard, 2021). This poor understanding of precipitation extends to localised extremes that result from the triggering of convective events by small-scale topographic features (Orr et al., 2017; Bhardwaj et al., 2021; Ren et al., 2017; Dimri et al., 2017; Thayyen et al., 2013; Potter et al., 2018), which are often associated with rapid hydrological responses as well as hydro-meteorological hazards such as floods and landslides (Qazi et al., 2019; Lutz et al., 2016; Ji et al., 2020; Dimri et al., 2017; Thayyen et al., 2013; Das et al., 2006).

One of the approaches to overcome the issues related to the limited gauge networks in HKH has been to develop daily gridded datasets with wide spatial and temporal coverage. These include gauge-based products such as the Asian Precipitation Highly Resolved Observational Data Towards Evaluation of Water Resources (APHRODITE; Yatagai et al., 2012), satellite-based products such as the Tropical Rainfall Measuring Mission (TRMM; Huffman et al., 2007), climate reanalysis products such as ECMWF Reanalysis v5 (ERA5; Hersbach et al., 2020), and multi-source products such as the Multi-Source Weighted-Ensemble Precipitation (MSWEP; Beck et al., 2019), which merges gauge, satellite, and reanalysis data. However, in HKH these datasets are characterised by large differences in both climatological and extreme precipitation values, with the lack of consensus confirming that our understanding of precipitation characteristics in this region is extremely poor (Bannister et al., 2019; Palazzi et al., 2013; Li et al., 2018). The large differences between these datasets are explained by the different types of observations used in them, as well as the methods used to compile them. For example, APHRODITE relies on distance-weighted interpolation of gauge values to derive precipitation patterns, which are difficult to robustly calculate in HKH due to the lack of in situ observations, as well as the large spatio-temporal precipitation gradients (Bannister et al., 2019; Ji et al.,

2020; Luo et al., 2020; Andermann et al., 2011). Recent alternatives such as MWSEP (Beck et al., 2019), which is arguably one of the best global precipitation datasets, also relies on gauge data and is therefore much less well constrained to observations in HKH compared to elsewhere. Consequently, alternative tools are needed to better understand the detailed spatio-temporal characteristics of precipitation in HKH.

Dynamical downscaling of coarse spatial resolution reanalysis datatsets (e.g., approximately 30 km for ERA5) using a regional climate model (RCM) is increasingly being used to produce high-resolution gridded precipitation products over HKH (Norris et al., 2020; Bannister et al., 2019; Maussion et al., 2011; Wang et al., 2021). These RCMs are characterised by spatial resolutions from 1-10 km that are generally able to resolve the complex terrain and thus better represent precipitation variability, and especially extremes. However, RCM outputs are still subject to errors and uncertainty (Giorgi, 2019), which can be exacerbated in mountain areas due to the complexity of the terrain (Sanjay et al., 2017; ul Hasson et al., 2019). For example, while reanalysis-driven RCM simulations are able to capture the large-scale circulation accurately (e.g., summer monsoon and westerly disturbances) by using either nudging or frequent initialisation techniques (Norris et al., 2020; Bannister et al., 2019; Maussion et al., 2011; Wang et al., 2021), they can still be subject to deficiencies in the representation of key physical processes such as the local valley wind regime, boundary layer, and cloud microphysics (Orr et al., 2017; Potter et al., 2018), as well as discrepancies between real and simulated orography (Eden et al., 2012). Statistical post-processing techniques, such as bias correction, are therefore often applied to improve the accuracy of RCM outputs, including precipitation (e.g., Shrestha et al., 2017; Bannister et al., 2019; Dimri, 2021; Tazi et al., 2023).

The model output statistics (MOS) approach to bias-correcting RCM simulations involves developing statistical relationships between RCM outputs, used as predictors, and observations, used as predictands (e.g., Klein and Glahn, 1974; Maraun and Widmann, 2018). MOS post-processing methods are usually deployed in either single-site or multi-site fashion to correct RCM simulations for locations where observations are available. However, in settings where gauge measurements are spatially sparse, MOS post-processing can also be used to adjust RCM precipitation output at ungauged locations (e.g., Samuel et al., 2012). In regions such as HKH where standard spatial interpolation techniques fail to capture the local-scale spatio-temporal precipitation variability, such an approach is fundamental; yet, it has not received much attention in the past. Bannister et al. (2019) applied bias correction to ungauged locations by using a deterministic, distribution-wise MOS method to adjust RCM precipitation outputs across two Himalayan basins. Additionally, MOS post-processing for bias correction can sometimes also involve downscaling RCM outputs to higher spatial resolutions (i.e. they correct for biases as well as downscale from a coarser to a finer scale). Hereafter, we use the term post-processing to refer to the combination of downscaling and bias-correcting.

Traditional MOS methods post-process the marginal distribution of the RCM output deterministically. In this setting, a specific set of predictors always yields the same corrected value and the spatio-temporal structure of the simulated output is not explicitly altered. This implicitly assumes that local-scale spatio-temporal variability is completely captured by the RCM-simulated gridbox variability. Whilst this assumption might hold in the case of pure bias correction, if the post-processing also involves downscaling to point observations (or higher-resolution gridded data) then deterministic approaches are not appropriate and a probabilistic method should be used instead (Maraun, 2013). Furthermore, using regression-based MOS methods, the synchrony (or pairwise correspondence) between reanalysis-driven RCMs and observations can be leveraged to

correct for biases in the temporal representation of RCM precipitation outputs, which can often be large (Lafon et al., 2013). In settings where the pairwise correspondence between RCM hindcasts and observations is low, such as in the HKH, probabilistic regression-based MOS methods can also provide value by characterising predictive uncertainty.

Previous studies have developed regression-based MOS methods based on artificial neural networks to statistically post-process precipitation (e.g., Cannon, 2008; Sachindra et al., 2018; Baño-Medina et al., 2020; Vaughan et al., 2021). For example, the multi-site precipitation downscaling framework proposed by Cannon (2008) employs artificial neural networks for probabilistic regression. More advanced regression-based MOS model architectures have also been leveraged recently to undertake this task, including convolutional neural networks (Baño-Medina et al., 2020), autoencoders (Vandal et al., 2019), and neural processes (Vaughan et al., 2021). However, all these methods generally rely on the availability of abundant training data, and thus focus on data-rich regions. The potential of regression-based MOS post-processing for ungauged mountain locations (such as in HKH) remains untapped.

In this pilot study, we introduce the Generalised Probabilistic Regression (GPR) MOS approach for post-processing (i.e., downscaling and bias-correcting) RCM daily precipitation outputs using sparse gauge data in HKH. This approach extends the pairwise stochastic MOS framework proposed by Wong et al. (2014) and leverages probabilistic regression models (i.e., generalised linear models and multi-layer perceptrons). The key advantage of the GPR approach is that is it capable of generalising to ungauged locations, whilst also capturing the uncertainty that arises both from this spatio-temporal generalisation and from the asynchronous timing of precipitation between RCM output and observations. Thus, using the GPR approach we can leverage a discrete and relatively sparse network of in situ observations to improve precipitation maps (i.e., gridded products) for HKH whilst quantifying the uncertainty of our estimates. The GPR approach can also be viewed as a probabilistic spatio-temporal interpolation technique for daily precipitation observations informed by (or conditioned on) RCM simulations and other contextual factors. Furthermore, the framework we employ is, in essence, a conditional MOS precipitation generator that is consistent with the RCM-simulated weather (Cannon, 2008; Wong et al., 2014).

We test the GPR framework by post-processing daily precipitation output from an RCM simulation of HKH produced using the Weather Research and Forecasting (WRF) model for three target regions, namely, the eastern and western reaches of the Upper Indus Basin, and the central part of the Upper Ganges Basin, which hereafter are referred to as East UIB, West UIB, and Central UGB respectively (Fig. 1). Together, these three regions span a wide portion of HKH and have very different characteristics in terms of geography, orography, climatology and observational network / data availability, i.e., providing a diverse range of conditions / challenges for the GPR framework in order to robustly test it. Finally, although the focus of this study is HKH, the results of this exercise should be applicable to other data-sparse mountain ranges in the world.

## 2 Data and Methods

### 2.1 Target regions and datasets

The West UIB region includes the Gilgit-Baltistan area, which is located in Karakoram and western Himalaya (Fig. 1). The Gilgit-Baltistan area is 72,971 km$^2$ in size, and includes the Hunza and Gilgit rivers, as well as the main branch of the Indus

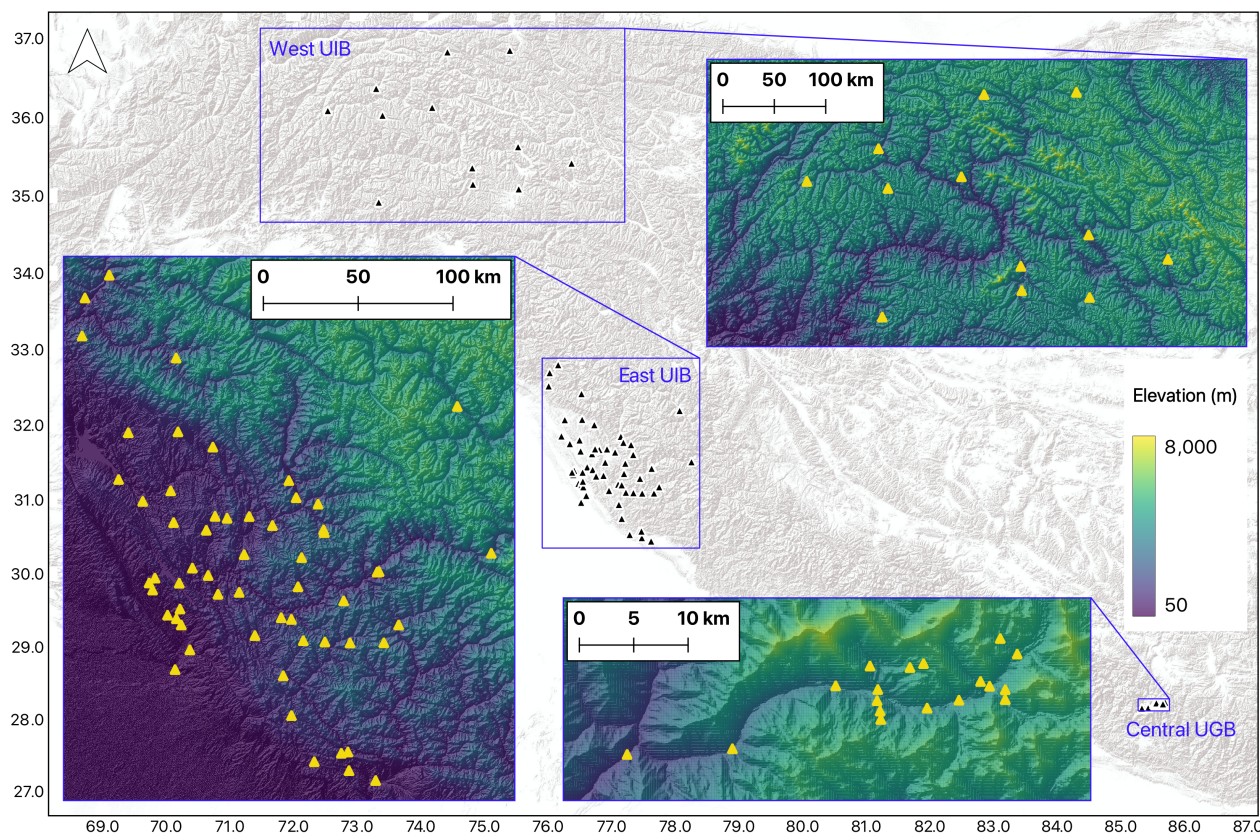

**Figure 1.** Map showing the three target regions across HKH: West Upper Indus Basin (West UIB), East Upper Indus Basin (East UIB), and Central Upper Ganges Basin (Central UGB), including the location of the gauge measurements (black triangles). Inset maps show more detail for each of the target regions, including the elevation of the topography (shading) and the location of gauge measurements (yellow triangles). The topography dataset shown in the inset maps is from the Shuttle Radar Topography Mission (SRTM) digital elevation model.

**Table 1.** Elevation range and summary of the stations used to provide daily precipitation observations for each of the three target regions. Note that each station contains gaps in the instrumental record.

| Region | Elevation range (m.a.s.l.) | No. stations | Period covered | No. datapoints |
|--------|---------------------------|--------------|----------------|----------------|
| West UIB | 1,460 - 4,707 | 12 | 1995-2014 | 76,860 |
| East UIB | 265 - 3,645 | 58 | 1980-2013 | 364,713 |
| Central UGB | 1,406 - 5,090 | 20 | 2012-2014 | 15,152 |

(Iqbal et al., 2019). For this study, the daily precipitation records available for this area consist of 12 stations over the period 1995-2015 (comprising a total of 76,860 data points), which range from 1,460 to 4,707 meters above sea level (m.a.s.l.) (Table 1). The average distance between neighbouring stations is around 60 km.

The East UIB region includes the Sutlej River basin and the Beas River basin, which are situated in western Himalaya. The Sutlej River basin has an area of 60,803 km$^2$ (above the Bhakra dam) and its river is the largest and easternmost of the tributaries of the Indus (Fig. 1). The Beas River basin has an area of 12,286 km$^2$ (above the Pong dam) and its river is itself a tributary of the Sutlej River. The available daily precipitation records in these basins (and neighbouring areas) come from 58 stations over the period 1980-2013 (364,713 data points), which range from 256 to 3,645 m.a.s.l. (Table 1). The majority

of the stations are located in the lower reaches of both catchments (Fig. 1), i.e., a significant part of the catchment area sits at elevations above the highest monitoring station. The average distance between neighbouring stations is around 15 km, which constitutes a relatively-dense network of precipitation measurements compared to West UIB, as well as many other areas in HKH (Nepal et al., 2023).

    The Central UGB region includes the Langtang River catchment, which is situated in central Himalaya (Fig. 1). It consists of

135 a relatively small area, and its river is a tributary to the Ganges (Fig. 1). The available daily precipitation records for this region come from 21 stations from 2012-2014 (15,152 data points), which range from 1,406 to 5,090 m.a.s.l. (Table 1). The average distance between neighbouring stations is less than 2 km, which makes this one of the most dense networks of precipitation measurements in HKH (Steiner et al., 2021; Shea et al., 2015).

    The WRF simulation is by Norris et al. (2019). It dynamically downscales 36 years of Climate Forecast System Reanalyses

data (Saha et al., 2010) from 1979 to 2015 over HKH at a spatial resolution of 6.7 km. We use multiple outputs (including daily precipitation) from this simulation from 1980 to 2014 that cover the three target regions (see Table 2). Norris et al. (2019) found that daily precipitation output from this simulation was better correlated with HKH gauge data in winter (correlation coefficient of 0.70) than in summer (correlation coefficient of 0.56). Additionally, over the Karakoram (West UIB), the simulated precipitation had a relatively substantial negative bias (Norris et al., 2017). Note that the station data described above and

the WRF precipitation output are independent, i.e., the data was not assimilated into the Climate Forecast System Reanalysis. Finally, the terrain elevation of the three target regions (and stations) is taken from the Shuttle Radar Topography Mission (SRTM) digital elevation model, which has a spatial resolution of 30 m.

## 2.2 Generalised Probabilistic Regression (GPR) approach to MOS

The GPR approach to MOS post-processing involves predicting the probability $p$ over daily precipitation $y$, conditional on a set of predictors $x$, using regression models with parameters $\phi$ and whose output variables characterise a stochastic process (Eq. 1). Importantly, we assume that the probability over daily precipitation at one spatio-temporal location is conditionally independent from all other spatio-temporal locations, and thus the joint conditional probability $p(y|x)$ factorises into the product of $p(y_m|x_m)$, where $m$ ranges from 1 to $M$ and denotes a spatio-temporal location (i.e., there is a unique $m(s,t)$ for every combination of spatial location $s$ and time $t$).

$$p_\phi(y|x) = p_\phi(y_1, y_2, ..., y_m | x_1, x_2, ..., x_m) = \prod_{m=1}^{M} p_\phi(y_m|x_m). \tag{1}$$

More concretely, the GPR approach uses regression models $f_\phi$ that map from inputs $\mathbf{x}_m$ to outputs $\theta_m$ (Eq. 2). The input vector $\mathbf{x}_m$ is $D$-dimensional, whereas the output vector $\theta_m$ is $N$-dimensional and it explicitly parametrises the conditional probability distribution over daily precipitation $y_m$. We use three regression model architectures, namely, vector generalised linear models (VGLMs; Song, 2007) and two fully-connected artificial neural networks, also referred to as multi-layer perceptrons (MLPs; Rumelhart et al., 1986).

$$\theta_m = f_\phi(\mathbf{x}_m). \tag{2}$$

In VGLMs, the mapping from inputs $\mathbf{x}_m$ to outputs $\theta_m$ in Eq. 2 involves two key transformations. Firstly, a linear transformation, parametrised by a matrix of weights $\mathbf{W}_1$, is applied to $\mathbf{x}_m$. Secondly, a non-linear transformation $\mathbf{g}$ is subsequently applied to obtain the output vector $\theta_m$. Note that each element $\theta_m^n$ is generated by applying a specific link function $g_n$, where $n$ ranges from 1 to $N$ and indexes each element of $\theta_m$ and $\mathbf{g}$. This element-wise non-linear transformation ensures that the resulting output values are valid parameters of the predicted probability distribution (Eq. 3).

$$\theta_m = \mathbf{g}(\mathbf{W}_1^\top \mathbf{x}_m). \tag{3}$$

Thus, for VGLMs, parameters $\phi = \mathbf{W}_1$.

In contrast, in MLPs the mapping from inputs $\mathbf{x}_m$ to outputs $\theta_m$ in Eq. 2 involves passing $\mathbf{x}_m$ through multiple hidden layers, in sequence. The mapping from each layer to the next layer involves several linear transformations (determined by the number of units in that layer and parametrised by matrices $\mathbf{W}_1, \mathbf{W}_2, ..., \mathbf{W}_{H+1}$, where $H$ is the number of layers), each followed by a non-linear activation function $a$ (Eq. 4). This structure allows MLPs to model more complex (non-linear) relationships than a linear model. We use a very small MLP with one hidden layer of 10 units, where $\phi = \{\mathbf{W}_1, \mathbf{W}_2\}$ (Eq. 4; hereafter referred to as MLP$_S$) and a larger version with two hidden layers of 50 units each, where $\phi = \{\mathbf{W}_1, \mathbf{W}_2, \mathbf{W}_3\}$ (Eq. 5; hereafter referred to as MLP$_L$). Rectified linear unit (ReLU) non-linearities are used as hidden layer activations $a$ in both MLP architectures, except for the last layer, which also employs a vector of link functions $\mathbf{g}$ to map each output variable.

$$\theta_m = \mathbf{g}(\mathbf{W}_2^\top a(\mathbf{W}_1^\top \mathbf{x}_m)). \tag{4}$$

**Table 2.** Summary of variables used as inputs to the GPR post-processing models, grouped by variable type.

| Description | Acronym | Units |
|---|---|---|
| WRF liquid and total precipitation | RAIN, PRECIP | mm/day |
| WRF water vapour path | WVP | kg/m$^2$ (daily average) |
| WRF convective available potential energy | CAPE | m$^2$/s$^2$ (daily average) |
| WRF temperature daily avg, max, min, and range at 2 m | $T_{2m}$, $T_{2m,MAX}$, $T_{2m,MIN}$, $T_{2m,R}$ | K |
| WRF zonal and meridional wind at 10 m, 500 hPa, and 250 hPa | $U_{10m}$, $V_{10m}$, $U_{500}$, $V_{500}$, $U_{250}$, $V_{250}$ | m/s (daily average) |
| WRF vertical wind at 500 hPa and 250 hPa | $W_{500}$, $W_{250}$ | m/s (daily average) |
| WRF relative humidity at 2 m and 500 hPa | $RH_{2m}$, $RH_{500}$ | % (daily average) |
| WRF orography (based on surface geopotential height) | GPH | m |
| WRF land use index | LU | - |
| Latitude, longitude (of target station) | Y, X | m |
| Terrain elevation (of target station) | Z | m |
| Day of year encoded via sine and cosine functions | $DoY_{SIN}$, $DoY_{COS}$ | - |
| Year | Year | - |

$$\theta_m = \mathbf{g}(\mathbf{W}_3^\top a(\mathbf{W}_2^\top a(\mathbf{W}_1^\top \mathbf{x}_m))). \tag{5}$$

In order to post-process WRF daily precipitation outputs, the three GPR model architectures use an input vector $\mathbf{x}_m$ that consists of $D = 26$ variables listed in Table 2, i.e., resulting in a 26-dimensional vector. This includes outputs from the WRF simulation at some spatio-temporal location $m$, as well as other context variables relating to the geographical location, orography, and date. The outputs from the WRF simulation include daily precipitation (i.e., the variable that is being post-processed), as well as other variables that are closely related to precipitation, cloud properties, and convective processes, such as convective available potential energy, cloud water vapour path, relative humidity, horizontal and vertical winds, and temperature.

To characterise the conditional probability over daily precipitation, we employ a Bernoulli-gamma mixture model, which is capable of jointly accounting for precipitation occurrence and magnitude and has been used in previous studies (Williams, 1998; Cannon, 2008). Precipitation occurrence is modelled by introducing a Bernoulli random variable $r_m$, which takes the value 1 with probability $\pi_m$ and the value 0 with probability 1-$\pi_m$. When $r_m = 1$, precipitation magnitude $y_m$ is modelled by a gamma distribution with parameters $\alpha \in (0,\infty)$ and $\beta \in (0,\infty)$. The Bernoulli-gamma mixture is implemented by specifying regression models architectures that generate an $N = 3$ dimensional output vector $\theta_m = [\pi_m, \alpha_m, \beta_m]$, using link functions $\mathbf{g} = [\text{sigmoid}(\cdot), \exp(\cdot), \exp(\cdot)]$. Further implementation details can be found in Appendix A.

## 2.3 Training, validation and testing

The GPR post-processing models are trained, validated, and tested using the daily precipitation observations $y^{\mathrm{obs}}$ (Table 1) as predictands, i.e., target values. We employ a k-fold cross-validation approach, which involves splitting the data by location (i.e., station) into $k$ folds, with $k-2$ folds being used for training, 1 fold for validating, and 1 fold for testing. This process is repeated $k$ times, ensuring each fold is used once for testing. During training, we optimise the model parameters $\phi$ to maximise the average log-likelihood of the training dataset. For this, we use stochastic gradient descent with a batch size of 128 and the Adam optimiser (Kingma and Ba, 2015) with an initial learning rate equal to $10^{-3}$. Here, we consider the validation step as part of the training process as it involves selecting, from the different model training iterations, the configuration of model parameters $\phi$ that maximises the average log-likelihood of the validation dataset, to avoid overfitting to the training data. We thus refer to the combined training and validation steps as training. Lastly, testing involves evaluating the performance of the trained models on the held-out locations in the test dataset.

## 2.4 Scaling factor approach

We compare the post-processed WRF precipitation results from the three GPR models against results using a widely-used deterministic MOS scaling factor approach (Maraun and Widmann, 2018), which we refer to as WRF$_{\mathrm{SF}}$. Here, the raw WRF daily precipitation output for station $s$ (hereinafter referred to as $y_s^{\mathrm{WRF}}$) is scaled by the ratio between total observed daily precipitation ($\sum_{m=1}^{M} y_m^{\mathrm{obs}}$) and total WRF-simulated daily precipitation ($\sum_{m=1}^{M} y_m^{\mathrm{WRF}}$), where data points indexed $m \in \{1,...,M\}$ correspond to locations other than $s$, to obtain $y_s^{\mathrm{WRF_{SF}}}$ (Eq. 6).

$$y_s^{\mathrm{WRF_{SF}}} = y_s^{\mathrm{WRF}} \left( \frac{\sum_{m=1}^{M} y_m^{\mathrm{obs}}}{\sum_{m=1}^{M} y_m^{\mathrm{WRF}}} \right). \tag{6}$$

The scaling factor method is also applied using a 10-fold spatial cross-validation approach, where the scaling factor is derived using $k = 9$ folds (i.e., data points indexed $m \in \{1,...,M\}$) and then applied to the data points in the remaining fold.

## 2.5 Experiments

We undertake four experiments that assess the performance of the three GPR post-processing models, as well as the scaling factor approach WRF$_{\mathrm{SF}}$. The four experiments represent increasingly complex (but realistic) ways of partitioning the available station data into subsets for training, validation, and testing, and are shown schematically in Fig. 2. In Experiment 1 (hereafter referred to as E1), we train separate GPR models for each region (i.e., separate-region models) and test them by post-processing the WRF precipitation output at held-out locations within that region. This experiment represents a baseline scenario, where models are trained and tested on the same region. In Experiment 2 (E2), we train GPR models using data from all three regions combined (i.e., a combined-region model) and test them by post-processing WRF precipitation output at held-out locations within each of the regions. This experiment therefore explores whether training a model over a diverse range of regions/settings and then applying it to each of these regions outperforms the separate-region (E1) models. Both E1 and E2 use

10-fold cross-validation. Experiment 3 (E3) is similar to E2 but trains and validates the models on combined data from two regions (consisting of 80% and 20% of the combined data, respectively), and tests on 100% of the data from a third, completely held-out, region. This experiment therefore explores whether a model that is trained over a set of regions/settings can generalise to an entirely different new region. In E3 we use 3-fold cross-validation to ensure that each region is held-out for testing once to produce predictions for that region. Lastly, Experiment 4 (E4) is analogous to the separate-region (E1) experiment but splits the data up for training, validation, and testing based on the elevation of the stations. Here, for each region the top 10% elevation stations are withheld for testing, the next 10% are used for validation, and the remaining 80% (i.e., lowest elevation) stations are used for training. This experiment therefore explores whether models trained on data from the lower reaches of catchments, where the majority of stations are located, are capable of generalising to much higher elevations that are typically ungauged. Note that E4 therefore does not involve k-fold cross-validation.

## 2.6 Evaluation metrics

To evaluate the post-processed precipitation distributions from each of the three GPR post-processing models in each of the three target regions, we employ three strictly proper scoring rules (Gneiting and Raftery, 2007), which are the negative log-likelihood (NLL), the continuous rank probability score (CRPS), and the Brier score (BS) – defined below. For the CRPS and the BS, we calculate their associated skill scores CRPSS and BSS, respectively (e.g., Angus et al., 2024). These measure the improvement relative to the CRPS and BS for the raw WRF precipitation output, which is considered as our baseline. The BSS metric is used to assess the ability of different post-processing methods to capture various precipitation thresholds (0, 1, 10, 30 and 50 mm/day) that span the spectrum of precipitation events, ranging from no precipitation to very extreme events. The frequency and total number of events exceeding these thresholds are included in Table B1 (Appendix B) showing that 10, 30 and 50 mm/day represent extreme precipitation events for which nevertheless some amount of observations are available (e.g., 0.33%, 2.95%, and 0.68% of the total number of events exceed the 30 mm/day threshold at West UIB, East UIB, and Central UGB, respectively), as opposed to higher thresholds such as 100 mm/day, which have less than 0.2% of observations, i.e., too low to justify use of this threshold. In addition, we use the threshold-weighted CRPS (twCRPS; with precipitation thresholds of 10 and 30 mm/day) to further assess the tails of the predictive distributions, by calculating the associated skill score twCRPSS (see Appendix B). The CRPSS, BSS and twCRPS metrics are also used to evaluate the skill of $WRF_{SF}$. To complement this, we also use the mean squared error (MSE) and mean absolute error (MAE) to compare the performance of the post-processed precipitation distributions reduced to their mean values against $WRF_{SF}$ outputs, by again calculating their associated skill scores MSESS and MAESS (see Appendix B).

Moreover, to assess the trade-off between goodness-of-fit and model complexity, we consider several information criteria. In particular, we compute the Akaike Information Criterion (AIC; Akaike, 1973), its small-sample corrected version (AICc; Sugiura, 1978), and the Kullback Information Criterion (KIC; Cavanaugh, 1999). These criteria, which consider both the log-likelihood of the model and the number of parameters in the model, are defined in Appendix C. While our model evaluation primarily relies on cross-validation and proper scoring rules, these criteria offer a complementary perspective on overall model quality.

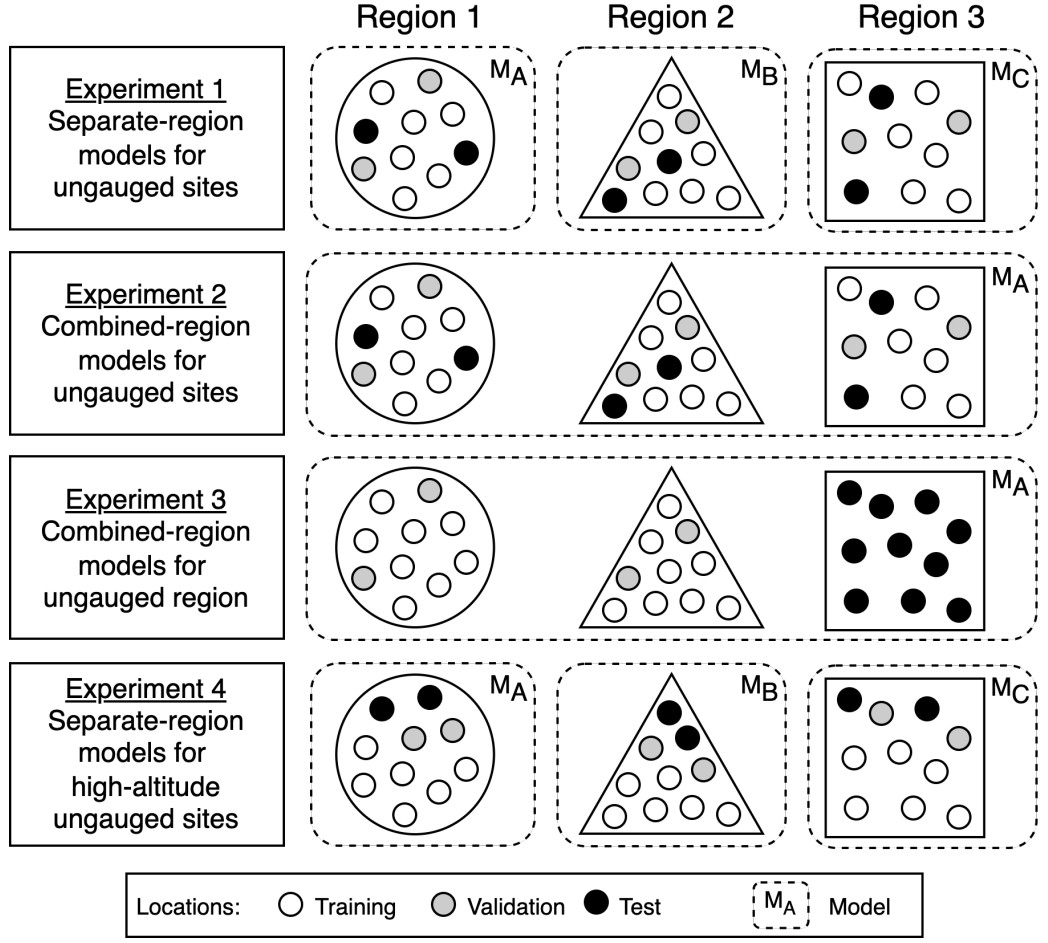

**Figure 2.** Schematic representation of the four experiments included in this study applied to three target regions. In Experiment 1, separate-region models for each of the three regions are trained and tested in held-out locations within each region. In Experiment 2, combined-region models are trained on data from all three regions and tested in held-out locations within those same regions. In Experiment 3, models are trained on two regions combined and tested in a third, completely held-out, region. Experiment 4 is analogous to Experiment 1 but held-out test locations are at higher elevations than those used for training and validation.

The NLL assesses the compatibility of probabilistic hindcasts with observed data, especially focusing on the probability of observed events under the predicted probability distributions. It is defined as the sum of the natural logarithms of the probability density function values at the observed data points (Eq. 7).

$$\text{NLL} = -\frac{1}{M} \sum_{m=1}^{M} \log(p(y_m^{\text{obs}}|x_m)), \tag{7}$$

where $M$ represents the number of observations, $y_m^{\text{obs}}$ denotes the $m$-th observed data point, and $p(y_m^{\text{obs}}|x_m)$ is the value of the predicted probability density function evaluated at that observed data point. NLL is the optimisation criterion used during training and therefore lower values indicate better performance.

The CRPS also measures how well probabilistic prediction matches the observed data $y_m^{\text{obs}}$ by measuring the distance between the predicted and observed cumulative distribution functions (CDFs). The CRPS is defined as the integral of the squared difference between the predicted CDF $F(y_m)$ and the observed empirical CDF, which is the Heaviside step function $H(y_m - y_m^{\text{obs}})$, over the entire range of possible values $y$ (Eq. 8).

$$\text{CRPS} = \frac{1}{M} \sum_{m=1}^{M} \int_{-\infty}^{\infty} (F(y_m) - H(y_m - y_m^{\text{obs}}))^2 dy_m. \tag{8}$$

Using this, the CRPSS is then calculated as:

$$\text{CRPSS} = 1 - \frac{\text{CRPS}}{\text{CRPS}_{\text{WRF}}}, \tag{9}$$

where $\text{CRPS}_{\text{WRF}}$ is the CRPS of the raw WRF precipitation output. Positive values of CRPSS indicate improved skill relative to the raw WRF output, with higher values indicating better performance. Note that for deterministic predictions (i.e., WRF and $\text{WRF}_{\text{SF}}$) the CRPS reduces to the mean absolute error between the predicted and observed values.

The BS (Wilks, 2006) measures the mean squared error between $M$ pairs of precipitation occurrence probabilities $\pi_m$ and binary observations $r_m^{\text{obs}}$ (Eq. 10), and allows for a detailed assessment of the predictive capacity across different levels of precipitation intensity.

$$\text{BS} = \frac{1}{M} \sum_{m=1}^{M} (\pi_m - r_m^{\text{obs}})^2. \tag{10}$$

Using this, the BSS is calculated as:

$$\text{BSS} = 1 - \frac{\text{BS}}{\text{BS}_{\text{WRF}}}, \tag{11}$$

where $\text{BS}_{\text{WRF}}$ is the BS of the raw WRF precipitation output. Positive BSS values indicate an improved skill relative to the raw WRF output, with higher values indicating better performance.

Finally, we extend the performance assessment for the combined-region models (E2) as these showcase the benefits and challenges associated with leveraging data from different regions. For this, we pool together the E2 held-out predictions for all (three) regions and use reliability diagrams and receiver operating characteristics (ROC) curves (e.g., Angus et al., 2024).

Reliability diagrams serve as a visual representation of the calibration accuracy of predicted probabilities for different precipitation levels (0, 1, 10 and 30 mm/day), extending the evaluation beyond pairwise-correspondence metrics such as NLL, CRPSS and BSS. Reliability diagrams display the relationship between predicted probabilities of precipitation exceeding a certain threshold and the actual observed frequencies, with a perfect agreement indicated by such relationship falling along the diagonal line on the graph. ROC curves offer an alternative perspective of probabilistic model performance across for different precipitation levels (0.1, 1, 10 and 30 mm/day). In particular, ROC curves assess the ability of probabilistic predictions to discriminate an event from a non-event by plotting the hit rate (i.e., ratio between the number of correctly predicted events and the total number of events) against the false alarm rate (i.e., ratio between the number of wrongly predicted events and the total number of events) using different predicted probability thresholds to transform the probabilistic prediction into a binary prediction of occurrence (Wilks, 2006; Angus et al., 2024). To allow for a better graphical differentiation of the ROC curves for different precipitation events, data points with no observed precipitation are excluded from this analysis.

## 2.7 Feature ablation

To determine the most influential input variables for the three GPR post-processing models, we perform a feature ablation analysis for E2 (Zeiler and Fergus, 2014; Kokhlikyan et al., 2020). Feature ablation is a technique that replaces each input variable (or feature) in $\mathbf{x}_m$ with a baseline value (in this case, zero) and measures the impact this has on the output vector $\theta_m$. This is done for each of the $D = 26$ input variables from Table 2 by running the trained GPR models 26 times, each time with a different input variable in $\mathbf{x}_m$ replaced by zero, thereby obtaining predictions $\theta_m$ for each ablated-feature input configuration. For each feature, the average of the absolute value of the differences between the ablated-feature model predictions and the original model predictions is computed for each output variable, i.e., $\pi$, $\alpha$ and $\beta$.

## 3 Results

Figure 3 shows how differences between observed and WRF-simulated mean daily precipitation vary by region and with terrain elevation. In West UIB and Central UGB, WRF systematically overestimates precipitation for all stations, i.e., for the full elevation range of the stations. This overestimate is especially apparent in Central UGB, with station measurements showing values of around 2 mm/day, whereas WRF generally simulates 8-10 mm/day. In East UIB, WRF underestimates precipitation at low-elevation stations (below around 1000 m.a.s.l.) and broadly overestimates it at higher elevations (especially above 2000 m.a.s.l.). These results generally show that the WRF output is characterised by highly variable precipitation biases across the three target regions that are consistent with complex elevation and hydro-climatological dependencies, and therefore that generalising MOS post-processing at ungauged locations is likely challenging. In addition, Figure B1 (Appendix B) shows the Pearson correlation coefficient between raw WRF-simulated and observed daily precipitation timeseries for the different stations, showing that while regional differences exist (e.g., correlations in Central UGB, where stations are relatively close to each other, are higher than in East UIB and West UIB, where the distance between is stations are larger) the pairwise correspondence (or synchrony) between WRF and observations is low, ranging between 0.1 and 0.5.

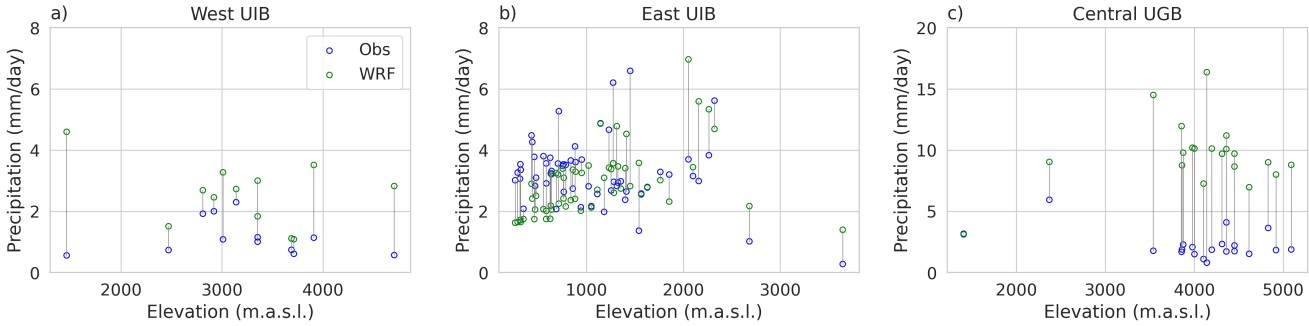

**Figure 3.** Difference between observed and WRF-simulated mean daily precipitation (mm/day) for each gauging station as a function of station elevation for a) West UIB, b) East UIB, and c) Central UGB.

Table 3 evaluates the performance of the three GPR post-processing models (VGLM, $MLP_S$ and $MLP_L$) for each experiment and region using NLL. For E1, which assesses separate-region models for each of the three target regions, both MLP models marginally outperform the VGLM model in all three regions, with $MLP_L$ performing best. Similar results are also apparent for E2, which tests how combined-region models generalise to held-out locations within those regions. Comparison of E2 and E1 shows that the combined-region GPR models (E2) perform marginally better than the separate-region GPR models (E1) for East UIB, but are marginally poorer for West UIB and Central UGB. However, the NLL values for E1 and E2 for each region and each GPR model are very similar and only differ in the second decimal place, i.e., there is little difference between the performance of either the separate-region or combined-region models, as well as between the three GPR models. For E3, which is analogous to E2 but the GPR models are trained using data from two regions combined together and tested on the third region, NLL values are much higher compared to E2, highlighting a considerable drop in performance in all three regions. For example, the NLL value for $MLP_L$ for West UIB is 1.24 for E2 and 2.46 for E3. Additionally, in E3, the range of NLL values within each region is relatively wide compared to E2, indicating that GPR performance in E3 is more sensitive to the choice of model architecture. For example, for West UIB the NLL value is 1.50 for VGLM, 2.84 for $MLP_S$, and 2.46 for $MLP_L$. For E4, which is analogous to E1 but testing happens at locations with higher elevations than those seen by the models during training, the performance of all GPR models slightly drops compared to E1. However, the NLL values for E1 and E4 for West UIB and East UIB still only differ in the second decimal place. Here, $MLP_S$ shows the best performance for West UIB and East UIB, while for Central UGB it is $MLP_L$.

In terms of differences across all experiments and models for each region, NLL values in East UIB are lower than those in West UIB by around 0.1, while NLL values in West UIB are in turn lower than those in Central UGB by (typically) 0.2 to 0.3 (Table 3). This pattern highlights the dominance of regional variability, driven by the quality/bias of the raw WRF output and the amount of station data available. For example, East UIB has the best-performing models and also the highest number of daily precipitation measurements (364,713), with West UIB having the second best-performing models and also the second

highest number of measurements (76,860), and finally Central UGB has the poorest performing models and also the least
number of measurements (15,152) (Table 1).

**Table 3.** NLL values of post-processed daily WRF precipitation for the three GPR model architectures (VGLM, $MLP_S$ and $MLP_L$), calculated for all four experiments and all three target regions. Lower NLL values indicate better performance, with the best-performing GPR model for each experiment and region shown in bold. NLL values are directly comparable across experiments and regions. Note that $MLP_L$ for E3 West UIB was trained using a learning rate of $10^{-4}$ to ensure training convergence, while the other experiments used $10^{-3}$.

| Experiment | West UIB | | | East UIB | | | Central UGB | | |
|---|---|---|---|---|---|---|---|---|---|
| | VGLM | $MLP_S$ | $MLP_L$ | VGLM | $MLP_S$ | $MLP_L$ | VGLM | $MLP_S$ | $MLP_L$ |
| E1 | 1.263 | 1.235 | **1.226** | 1.179 | 1.159 | **1.150** | 1.480 | 1.437 | **1.420** |
| E2 | 1.267 | 1.245 | **1.240** | 1.181 | 1.156 | **1.144** | 1.501 | 1.477 | **1.448** |
| E3 | **1.502** | 2.835 | 2.457 | 2.414 | 1.531 | **1.483** | **1.683** | 1.985 | 1.950 |
| E4 | 1.283 | **1.250** | 1.251 | 1.187 | **1.160** | 1.173 | 1.875 | 1.553 | **1.511** |

Table 4 extends the performance assessment by showing the CRPSS for the three GPR models. The CRPSS values are largely consistent with the NLL results. For example, both MLP models outperform VGLM in all three regions for E1 and E2, and $MLP_L$ is the best overall performing model. The CRPSS values for E1 and E2 for each region are very similar and only differ in the second decimal place, which was also found for the NLL results. For E3, all three GPR models exhibit lower CRPSS values and larger differences between models compared to E2, i.e, consistent with the considerable performance drop in all three regions and a higher sensitivity to the choice of model architecture found by NLL results. However, as CRPSS values for the GPR models are relative to the reference $CRPS_{WRF}$, the positive CRPSS values achieved still represent an improvement in skill relative to WRF. For E3, $MLP_S$ performs best for West UIB and East UIB (in contrast to VGLM for West UIB and $MLP_L$ for East UIB for the NLL results), while VGLM performs best for Central UGB (in agreement with the NLL results). For E4, the performance of all models slightly drops compared to E1, with the $MLP_L$ model still performing best across all three regions (in contrast to $MLP_S$ for West UIB and East UIB for the NLL results, but in agreement with $MLP_L$ for Central UGB, for the NLL results). CRPSS values also display differences across all experiments and models for each region, with the highest values of around ~0.8 in Central UGB, followed by values of ~0.6 for West UIB, and finally ~0.4 for East UIB. However, as CRPSS values are relative to (and thus influenced by) $CRPS_{WRF}$, which varies for each region, they are therefore not directly comparable across regions.

Table 4 also shows CRPSS values for $WRF_{SF}$. For all experiments and regions, the CRPSS values for $WRF_{SF}$ are lower than those for the GPR models, indicating that the performance of the GPR models is superior to $WRF_{SF}$. For example, for E1 and E4 the $WRF_{SF}$ values for East UIB are close to zero (-0.07), but positive in West UIB (~0.4) and Central UGB (0.69), i.e., indicating negligible improvement relative to the raw WRF output for East UIB, but some improvement for West UIB and Central UGB. This is likely related to the direction of WRF-simulated biases changing with elevation for East UIB, while for West UIB and Central UGB the biases are larger unidirectional (Fig. 3), i.e., large and unidirectional biases are more easily

post-processed and thus the scaling factor approach is also effective. For E2, $WRF_{SF}$ exhibits CRPSS values close to zero, indicating negligible improvement relative to the raw WRF output. For E3, $WRF_{SF}$ shows values close to zero for West UIB (-0.03) and Central UGB (-0.04), but positive values for East UIB (0.35). Additionally, Tables B2 and B3 (Appendix B) show MSESS and MAESS values, respectively, for all three GPR models as well as $WRF_{SF}$. The results yielded by these metrics are broadly in line with those for CRPSS, and confirm that while the relative performance of GPR models varies depending on the assessment metric, the best-performing GPR models still outperform $WRF_{SF}$ for all experiments and regions (except for E3 in East UIB, where $MLP_S$ and $WRF_{SF}$ yield the same MAESS value).

**Table 4.** CRPSS values of post-processed daily WRF precipitation for the three GPR model architectures (VGLM, $MLP_S$ and $MLP_L$) and $WRF_{SF}$, calculated for all three regions and all four experiments. Higher CRPSS values indicate better performance, with the best-performing MOS method for each experiment and region shown in bold. Positive CRPSS values indicate improved skill relative to raw WRF hindcasts. CRPSS values are directly comparable across experiments but not across regions. Note that $MLP_L$ for E3 West UIB was trained using a learning rate of $10^{-4}$ to ensure training convergence, while the other experiments used $10^{-3}$.

| Experiment | West UIB | | | | East UIB | | | | Central UGB | | | |
|---|---|---|---|---|---|---|---|---|---|---|---|---|
| | VGLM | $MLP_S$ | $MLP_L$ | $WRF_{SF}$ | VGLM | $MLP_S$ | $MLP_L$ | $WRF_{SF}$ | VGLM | $MLP_S$ | $MLP_L$ | $WRF_{SF}$ |
| E1 | 0.646 | 0.654 | **0.657** | 0.397 | 0.416 | 0.433 | **0.434** | -0.075 | 0.820 | 0.823 | **0.824** | 0.691 |
| E2 | 0.645 | 0.647 | **0.652** | 0.021 | 0.416 | 0.428 | **0.439** | 0.022 | 0.814 | 0.811 | **0.825** | 0.043 |
| E3 | 0.491 | **0.585** | 0.556 | -0.027 | 0.346 | **0.384** | 0.354 | 0.234 | **0.812** | 0.779 | 0.782 | -0.039 |
| E4 | 0.625 | 0.628 | **0.644** | 0.398 | 0.414 | 0.427 | **0.430** | -0.073 | 0.808 | 0.818 | **0.818** | 0.690 |

To better understand the performance of the GPR models for various precipitation intensities, Table 5 shows the BSS for the three GPR models for different daily precipitation thresholds. As expected, BSS values are consistent with CRPSS results but provide further granularity. For E1, E2 and E4, $MLP_L$ is best at capturing the probability over low-to-moderate precipitation thresholds (i.e., 0, 1 and 10 mm/day), whereas results for higher precipitation events (i.e., 30 and 50 mm/day) are variable. However, for each region and threshold the BSS values for different models generally only differ in the second decimal place, indicating that the performance of all GPR models is broadly similar for each region and threshold. For a threshold of 50 mm/day, smaller models generally perform marginally better for E1 (i.e., $MLP_S$ in West UIB and East UIB, and VGLM in Central UGB), E3 ($MLP_S$), and E4 (VGLM), whereas $MLP_L$ performs best in E2 for East UIB and Central UGB. For E3, the VGLM model considerably outperforms the MLP models for low precipitation thresholds (i.e., 0 and 1 mm/day) in West UIB and Central UGB, whereas $MLP_S$ performs best for higher thresholds (i.e., 30 and 50 mm/day) in these regions. Moreover, E3 also shows a much wider difference between models for West UIB and Central UGB. For example, for West UIB and a threshold of 30 mm/day, the BSS value is 0.47 for VGLM, 0.74 for $MLP_S$, and 0.68 for $MLP_S$.

For all experiments and regions, the BSS values for $WRF_{SF}$ are generally much lower than for the GPR models, indicating that the performance of the GPR models is superior to $WRF_{SF}$ (i.e., consistent with the CRPSS results). However, for E1 and E4, $WRF_{SF}$ can have BSS values that are comparable to the GPR models for higher thresholds, especially for East UIB and

Central UGB. For example, for Central UGB the BSS value for a threshold of 50 mm/day is 0.91 for $MLP_L$ and 0.90 for $WRF_{SF}$. In addition, Tables B4 and B5 (Appendix B) include twCRPSS values using 10 and 30 mm/day thresholds, respectively, for the three GPR models and $WRF_{SF}$. These results, which provide further insight regarding the ability of post-processing methods to capture extreme precipitation events, show that MLP models are best at characterising the tails of the predictive distributions in E1, E2, and E4 (excluding E2 for Central UGB in Table B5, where $WRF_{SF}$ outperforms GPR models), whereas VGLM is the most robust option in E3. In West UIB and Central UGB, twCRPSS values for thresholds of 30 mm/day show all methods performing poorly; this can be explained by the relative lack of observations for events exceeding this threshold in both regions (Table B1) .

Tables C3, C4 and C5 show how the three GPR models rank for each experiment and region, in terms of AIC, AICc and KIC information criteria, respectively. The rankings yielded by these metrics are generally consistent with the NLL results (Table 3), i.e., showing that MLP models tend to outperform VGLM models. These criteria penalise models with larger number of parameters (i.e., $MLP_L$; Table C2), especially in regions where the number of test data points is low (e.g., Central UGB and, to a lesser extent, West UIB; Table C1). AIC (Table C3) is the most lenient criteria, whereas AICc (Table C4) is more stringent with larger models and KIC (Table C5) penalises large models the most. Thus, in contrast to NLL results, $MLP_S$ is favoured over $MLP_L$; except in East UIB, where $MLP_L$ ranks first for E2 (based on AIC, AICc and KIC) and E3 (based on AIC and AICc).

Figure 4 shows reliability diagrams and corresponding observed precipitation event histograms for daily precipitation thresholds exceeding 0, 1, 10 and 30 mm/day for E2, i.e., the combined-region model. For low precipitation thresholds (i.e., 0 and 1 mm/day) the reliability diagrams show that the majority of predicted probabilities are well-calibrated (Fig. 4(a,b)), which is indicated by the data points corresponding analogous levels to predicted probability and observed frequency of precipitation (dashed line) coinciding with the diagonal line (solid line). However, the calibration accuracy declines for predicted probability values of between 0.9 and 1.0 due to the model overpredicting the observed frequency. This occurs when the count of predicted instances for a given cumulative probability value decreases below a threshold of around $10^3$ (Fig. 4(e,f)). This effect is even more evident for higher precipitation thresholds (i.e., 10 and 30 mm/day), where predicted probabilities and observed frequencies start to deviate at around 0.5 and 0.2, respectively (Fig. 4(c,d)), due to the model overpredicting the observed frequency. This also coincides with the number of predicted instances for these higher precipitation events dropping below a threshold of around $10^3$ (Fig. 4(g,h)), which highlights the challenge of predicting extreme precipitation events.

Figure 5 displays ROC curves for daily precipitation thresholds exceeding 0.1, 1, 10 and 30 mm/day for E2. This shows that GPR models exhibit superior binary classification accuracy compared to raw WRF and $WRF_{SF}$ for all precipitation thresholds. This is graphically depicted in Fig. 5 by the (probabilistic) curves sitting considerably above the (deterministic) point values in the diagrams, indicating that the GPR models have higher true hit rate at equivalent or lower false alarm rates compared to WRF and $WRF_{SF}$. Such a consistent pattern reinforces the evidence that GPR models not only provide a more nuanced forecast by quantifying uncertainty but also deliver a more reliable prediction in terms of discriminating between events and non-events for a given daily precipitation threshold. Additionally, Fig. 5 also shows for all precipitation thresholds that $MLP_L$

**Table 5.** BSS values of post-processed daily WRF precipitation for the three GPR model architectures (VGLM, $MLP_S$ and $MLP_L$) and $WRF_{SF}$, calculated for all three regions and all four experiments using a range of daily precipitation thresholds (0, 1, 10, 30 and 50 mm/day). Higher BSS indicate better performance, with the best-performing MOS method for each experiment and region shown in bold. Positive BSS values indicate improved skill relative to raw WRF hindcasts. BSS values are directly comparable across experiments but not across regions.

| E | T (mm) | West UIB | | | | East UIB | | | | Central UGB | | | |
|---|---|---|---|---|---|---|---|---|---|---|---|---|---|
| | | VGLM | $MLP_S$ | $MLP_L$ | $WRF_{SF}$ | VGLM | $MLP_S$ | $MLP_L$ | $WRF_{SF}$ | VGLM | $MLP_S$ | $MLP_L$ | $WRF_{SF}$ |
| E1 | 0 | 0.528 | 0.544 | **0.550** | 0.000 | 0.520 | 0.548 | **0.559** | 0.000 | 0.515 | 0.522 | **0.537** | 0.000 |
| E1 | 1 | 0.484 | 0.502 | **0.511** | 0.110 | 0.390 | 0.426 | **0.441** | -0.010 | 0.582 | 0.595 | **0.615** | 0.142 |
| E1 | 10 | 0.748 | **0.753** | **0.753** | 0.554 | 0.403 | 0.425 | **0.428** | -0.035 | 0.823 | 0.829 | **0.830** | 0.699 |
| E1 | 30 | 0.746 | **0.752** | 0.744 | 0.667 | 0.448 | **0.456** | 0.451 | -0.111 | **0.934** | 0.932 | 0.926 | 0.929 |
| E1 | 50 | 0.686 | **0.692** | 0.680 | 0.667 | 0.461 | **0.465** | 0.457 | -0.143 | **0.914** | 0.911 | 0.907 | 0.903 |
| E2 | 0 | 0.528 | 0.529 | **0.538** | 0.000 | 0.524 | 0.550 | **0.561** | 0.000 | 0.474 | 0.497 | **0.522** | 0.000 |
| E2 | 1 | 0.493 | 0.492 | **0.496** | 0.004 | 0.397 | 0.426 | **0.441** | 0.005 | 0.545 | 0.560 | **0.597** | 0.006 |
| E2 | 10 | 0.747 | 0.748 | **0.752** | 0.024 | 0.404 | 0.419 | **0.433** | 0.012 | 0.820 | 0.810 | **0.825** | 0.020 |
| E2 | 30 | 0.741 | **0.750** | 0.748 | 0.054 | 0.443 | 0.450 | **0.458** | 0.027 | **0.938** | 0.937 | **0.938** | 0.086 |
| E2 | 50 | 0.679 | **0.692** | 0.672 | 0.060 | 0.457 | 0.461 | **0.465** | 0.0048 | 0.917 | 0.917 | **0.919** | 0.101 |
| E3 | 0 | **0.519** | 0.299 | 0.308 | 0.000 | 0.427 | **0.458** | 0.442 | 0.000 | **0.418** | 0.051 | 0.117 | 0.000 |
| E3 | 1 | **0.485** | 0.331 | 0.338 | -0.005 | 0.263 | **0.340** | 0.314 | 0.054 | **0.534** | 0.318 | 0.360 | -0.003 |
| E3 | 10 | 0.471 | **0.741** | 0.728 | -0.028 | 0.326 | **0.375** | 0.329 | 0,172 | **0.817** | 0.811 | 0.808 | -0.024 |
| E3 | 30 | 0.473 | **0.745** | 0.680 | -0.060 | 0.400 | **0.416** | 0.399 | 0.341 | 0.934 | **0.939** | 0.936 | -0.074 |
| E3 | 50 | 0.516 | **0.686** | 0.483 | -0.090 | 0.432 | **0.438** | 0.431 | 0.410 | **0.916** | **0.916** | 0.912 | -0.099 |
| E4 | 0 | 0.521 | 0.534 | **0.546** | 0.000 | 0.519 | 0.544 | **0.555** | 0.000 | 0.441 | 0.461 | **0.501** | 0.000 |
| E4 | 1 | 0.458 | 0.486 | **0.504** | 0.110 | 0.390 | 0.422 | **0.432** | -0.010 | 0.566 | 0.556 | **0.577** | 0.142 |
| E4 | 10 | 0.732 | 0.720 | **0.736** | 0.554 | 0.400 | 0.418 | **0.422** | -.0.035 | 0.806 | 0.824 | **0.826** | 0.688 |
| E4 | 30 | 0.741 | 0.728 | **0.743** | 0.667 | 0.444 | 0.448 | **0.449** | -0.111 | 0.938 | **0.939** | 0.932 | 0.926 |
| E4 | 50 | **0.683** | 0.669 | **0.683** | 0.667 | **0.460** | 0.458 | 0.459 | -0.143 | **0.916** | **0.916** | 0.908 | 0.903 |

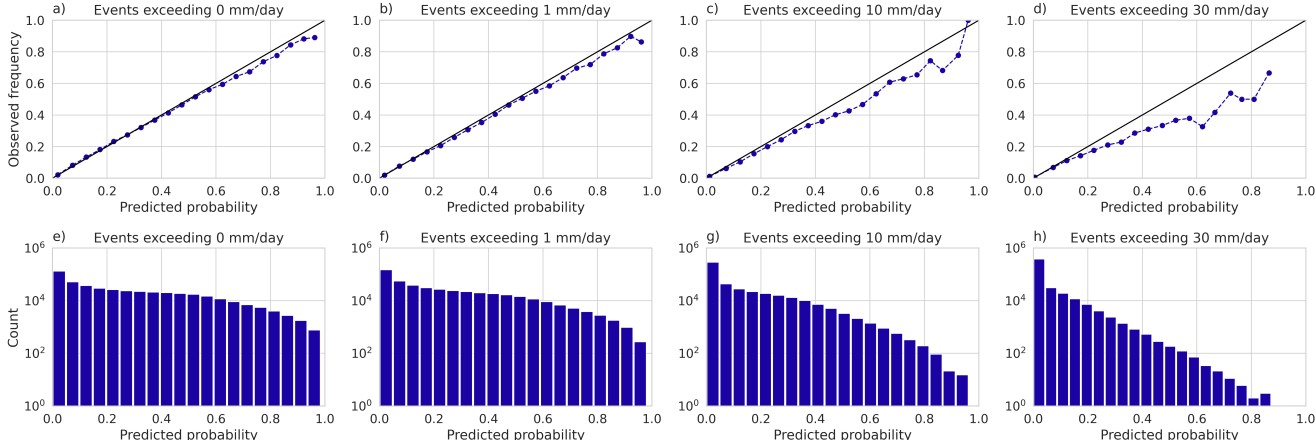

**Figure 4.** Reliability diagrams (top panels) and predicted probability histograms (bottom panels) for different observed daily precipitation thresholds of 0.1 mm/day (a,e), 1 mm/day (b,f), 10 mm/day (c,g), and 30 mm/day (d,h). Reliability diagrams (a,b,c,d) display the relationship between predicted probabilities and the actual observed frequencies of precipitation exceeding a certain threshold (dotted line), with a perfect agreement indicated by the diagonal line (solid line). Predicted probability histograms (e,f,g,h) display the counts of observed events exceeding a certain threshold associated with various predicted probability levels.

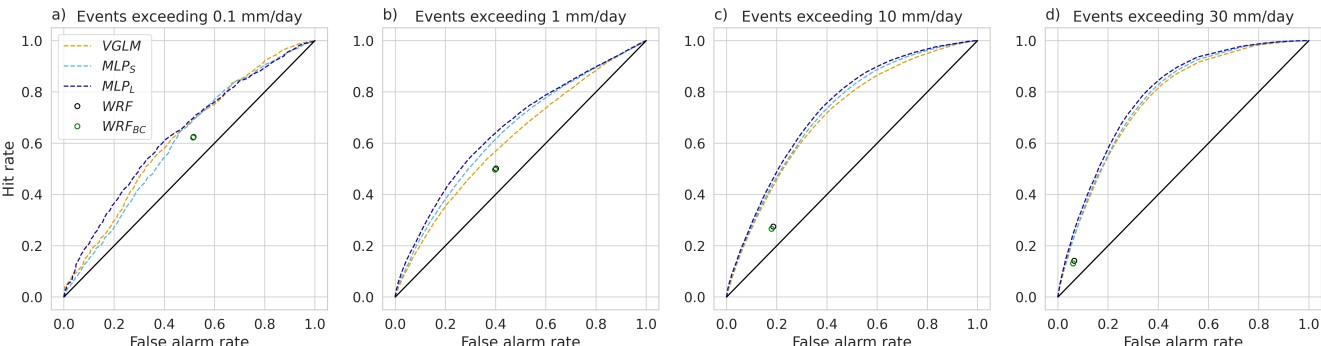

**Figure 5.** Receiver Operating Characteristic (ROC) curves yielded by the three GPR models, as well as hit rate and false alarm rate for WRF and WRF$_{SF}$, for different daily precipitation thresholds of 0.1 mm/day (a), 1 mm/day (b), 10 mm/day (c), and 30 mm/day (d). ROC curves exclude data points with no observed precipitation.

consistently yields the best performance of the GPR models. Note that ROC diagrams in Fig. 5 exclude data points with no observed precipitation (i.e., dry days) from the analysis.

Figure 6 shows the effect that progressively adding input variables (listed in Table 2) has on GPR model performance for E2, in particular on the held-out NLL value for MLP$_L$. Starting with a set of core input variables (i.e., PRECIP, RAIN, DOY$_{SIN}$, DOY$_{COS}$ and Z), the MLP$_L$ model yields a held-out NLL value of 1.33 (configuration labelled 'Core variables' in Fig. 6). By comparison, adding the spatial variables LAT and LON to the core set yields a held-out NLL value of 1.24 (labelled '+ Spatial

| Configuration | Additional input variables | Test NLL |
|---|---|---|
| Core variables | PRECIP, RAIN, DOY$_{SIN}$, DOY$_{COS}$, Z | 1.325 |
| + Spatial variables | LAT, LON | 1.237 |
| + Integrated variables | CAPE, WVP | 1.202 |
| + Temperature variables | T$_{2m}$, T$_{2m,MAX}$, T$_{2m,MIN}$, T$_{2m,R}$ | 1.197 |
| + Wind variables | U$_{10m}$, V$_{10m}$, U$_{500}$, V$_{500}$, U$_{250}$, V$_{250}$, W$_{500}$, W$_{250}$ | 1.195 |
| + Relative humidity vars. | RH$_{2m}$, RH$_{500}$ | 1.181 |
| + All other variables | GPH, LU, YEAR | 1.179 |

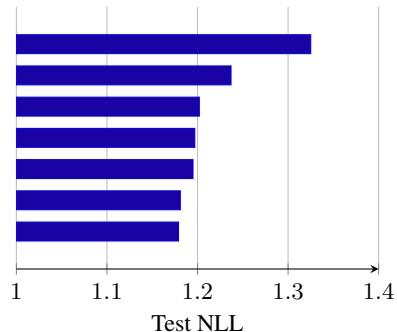

**Figure 6.** Test NLL yielded by MLP$_L$ in E2 for different configurations of input variables. Each configuration includes all the input variables from the preceding rows in addition to the specific set of input variables listed for that configuration. For example, the configuration labelled 'Core variables' contains the input variables PRECIP, RAIN, DOY$_{SIN}$, DOY$_{COS}$ and Z, whilst the configuration labelled '+ Spatial variables' contains 'Core variables' plus LAT and LON.

variables'), while further adding the vertically-integrated thermodynamic/cloud related input variables WVP and CAPE to this configuration yields a value of 1.20 (labelled '+ Integrated variables'), i.e, indicating a systematic improvement in skill as the number of input variables increases. However, adding the temperature-related input variables T$_{2m}$, T$_{2m,MAX}$, T$_{2m,MIN}$ and T$_{2m,R}$ (labelled '+ Temperature variables'), and then further adding the horizontal and vertical wind-related input variables U$_{10m}$, V$_{10m}$, U$_{500}$, V$_{500}$, U$_{250}$, V$_{250}$, W$_{500}$, and W$_{250}$ (labelled '+ Wind variables'), results in held-out NLL values of 1.20, i.e., indicating negligible gain in skill compared to the '+ Integrated variables' configuration. By contrast, adding the relative humidity variables RH$_{2m}$ and RH$_{500}$ further reduces the held-out NLL value to 1.18, which is likely due to these input variables being thermodynamic/cloud related. Lastly, adding the remaining input variables GPH (i.e., WRF orography), LU and YEAR yields a held-out NLL value of 1.18, i.e., no significant improvement.

Figure 7 assesses the relative influence that the input variables listed in Table 2 have on predicted outputs (i.e., the distributional parameters $\theta$, $\alpha$ and $\beta$) for the three GPR model architectures in E2. The feature importance analysis shows that the VGLM model is heavily influenced by a relatively limited set of input variables, whereas the MLP models (in particular MLP$_L$) leverage a more extensive array of predictors. The set of influential input variables is moderately consistent for all three output parameters. However, one notable exception is the dominant effect that T$_{2m}$ has on $\pi$ (Fig. 7a), but not on $\alpha$ and $\beta$ (Fig. 7(b,c)), for the VGLM model. The results further show that, for all three GPR models, the vertically-integrated thermodynamic/cloud variables (CVW and CAPE), the relative humidity variables (RH$_{2m}$ and RH$_{500}$), and the near-surface temperature variables (T$_{2m}$, T$_{2m,MAX}$, T$_{2m,MIN}$ and T$_{2m,R}$), stand out as important features, as well as inputs such as LAT, Z, and GPH. Moreover, and perhaps surprisingly, the precipitation input variables PRECIP and RAIN exhibit minimal influence on the GPR model outputs, which likely also explains the importance of the vertically-integrated thermodynamic/cloud variables and the relative humidity variables, as these play a dominant role in controlling precipitation. Furthermore, the contribution of the horizontal and vertical wind velocity fields to the GPR outputs is also relatively minor. It is important to note that this feature ablation analysis does

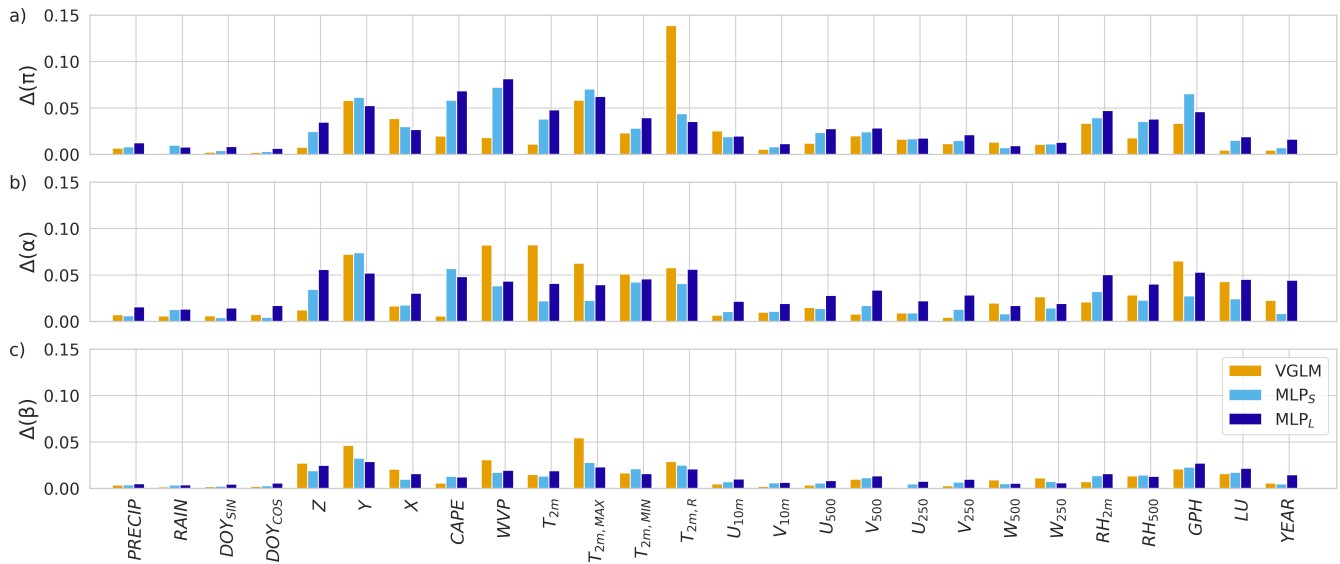

**Figure 7.** Feature ablation analysis for the three GPR models for E2, showing the effect that removing each input variable has on the output distributional parameters $\pi$ (a), $\alpha$ (b), and $\beta$ (c). Each vertical bar shows the average absolute-value output shift caused by ablating (i.e., replacing by zeros) a specific input variable, i.e., the larger the bar the more important the input variable is to the model.

not assess multivariate effects, but only the effect that removing a single input variable (i.e., replacing it with zeros) has on the GPR model outputs. As a result, the lack of influence shown by certain variables could be due to these being redundant given all other input variables (as evident in Fig. 6).

## 4   Discussion

### 4.1   Performance of GPR method for post-processing daily precipitation

In this study, we have shown that using a GPR approach to MOS post-process (i.e., downscaling and bias-correcting) daily precipitation output from a reanalysis-driven RCM, for which simulated and observed daily precipitation are expected to exhibit pairwise correspondence, improves predictions at ungauged locations across all tested regions, precipitation thresholds, and experiments. We use three scoring metrics (NLL, CRPSS and BSS) to evaluate the quality of hindcasts and find that, overall, the three GPR models we employ (VGLM, MLP$_S$, and MLP$_L$) exhibit similar performance (Tables 3, 4, 5) and consistently yield superior skill relative to the raw RCM output (WRF) and deterministic MOS bias correction (WRF$_{SF}$) (Tables 4, 5). We find that NLL (Table 3) and CRPSS (Table 4) yield similar relative rankings of GPR model performance. This is not surprising given that both the NLL and CRPS are strictly proper scoring rules that assess the goodness of fit of a predictive distribution against, in our case, a single observation, and have their minimums at the same value. However, the NLL is much more sensitive to extreme cases (as it involves a harsh penalty for events with low predicted probabilities) than the CRPS (Gneiting and Raftery,

2007). Therefore, it is insightful to corroborate the consistency of performance by using both scoring rules, whilst also noting that using the CRPS as the optimisation criterion for the GPR model parameters would have resulted in different GPR models and associated predictions. The BS assesses the accuracy of probabilistic predictions specifically for binary events. Whilst the CRPS is the integral of the BS over all real-valued probability thresholds and can therefore be viewed as a generalisation of the latter (Gneiting and Raftery, 2007), calculating the BS for specific precipitation thresholds provides additional granularity and shows insightful patterns for extreme precipitation events (Table 5).

MOS post-processing of RCM daily precipitation outputs at ungauged locations involves using separate groups of stations for model training, validation and testing. However, given the high spatial variability that daily precipitation exhibits both within and across regions, such an approach involves testing on out-of-distribution data, that is, models are assessed on their ability to generalise to held-out data that significantly deviates from the training data. This presents a particular challenge to representing extreme precipitation events. Daily precipitation in the three target regions is heavily skewed towards dry and very-low precipitation amounts, with high-intensity precipitation events accounting for a very low fraction of the data (not shown). This imbalance hinders the models' ability to learn a robust representation of the probabilities associated with extreme precipitation events, and how these vary spatially and temporally. For instance, the smaller / simpler GPR models such as VGLM and $MLP_S$ stand out as more robust options for extreme precipitation events (Table 5), which is likely because they have fewer trainable parameters (Table C2), and are thus able to learn less intricate patterns from the training data. In this context, information criteria such as AIC, AICc, and KIC (Tables C3, C4, and C5) emerge as complementary measures for comparing model quality (by balancing model performance and complexity); however, their interpretation for likely over-parameterised models (such as MLPs) is less straightforward because the effective number of parameters may be smaller than the nominal parameter count. In addition, given the variability of test set sizes across regions and experiments (Table C1), it is not clear which of these criteria is more appropriate, as each one of them has its own limitations and yields slightly different rankings. Therefore, we choose to interpret them with caution.

The four experiments we perform in this study assess GPR model performance across different post-processing tasks, which involve different ways of splitting data into training, validation and test sets. The dependencies between the training, validation and test datasets vary widely depending on whether the task involves a single region (E1) or multiple combined regions (E2), or extrapolating to either new regions (E3) or high elevations (E4). This explains the need for training models specific to each experiment and also highlights the spread of post-processing tasks considered in this work, attempting various degrees of generalisation. Consequently, the properties of the models trained to perform each task will also be different, which explains why GPR models ranked differently depending on the experiment.

In comparing GPR model performance across experiments, we showed that combined-region GPR models (E2) result in marginally better predictions than separate-region models (E1) for East UIB (Tables 3 and 4). Here, we hypothesise that East UIB benefits from combining data from all three regions because of the inherent challenges of this region, which is characterised by a complex bias-elevation distribution (Fig. 3), as well as an under-representation of station data at high elevations bands (>2,500 m.a.s.l., Fig. 1). We also find that GPR models are capable of improving daily precipitation hindcasts in completely ungauged regions (E3) by leveraging data from other regions (Tables 4 and 5). This result contradicts with the

assumption that because RCM daily precipitation biases are region-specific (Maussion et al., 2011; Norris et al., 2017; Bannister et al., 2019; ul Hasson et al., 2019), they are therefore not easily-transferable to other regions. It is likely that GPR models are partially able to overcome this issue because, conditional on enough information (i.e., relevant input features), daily precipitation biases for different regions share some similarities. E4 explores GPR model generalisation to locations situated at higher elevations than the gauging stations used to train the models. We find that GPR models successfully post-process WRF precipitation for all regions and thresholds, exhibiting only a marginal performance drop relative to E1 (Tables 3, 4, 5). This findinzg is particularly important given that much of the high elevation regions of HKH suffer from a profound lack of gauges (Pritchard, 2021; Thornton et al., 2022; Krishnan et al., 2019).

We find that training GPR models with progressively richer predictor configurations (i.e., with additional input variables, see Table 2) has a consistently positive effect on model performance, with some input variables (e.g., spatial, vertically-integrated, and relative humidity variables) driving most of the performance improvement (Fig. 6). Notwithstanding this, the performance gain yielded by incorporating additional input variables is strongly dependent on the previously added input variables. Thus, the relatively minor beneficial effect of variables such as temperatures and winds is likely only highlighting some degree of redundancy in the signal provided by different input variables. Furthermore, the three GPR models respond differently to the ablation of single input variables (Fig. 7), which highlights the complex interplay between input variables and model architecture. The reliance of the VGLM model architecture on a few input features contrasts the broader utilisation of inputs necessary for the $MLP_L$ model. The influence exhibited by input variables such as elevation and latitude (i.e., representing topography), and convective available potential energy (CAPE), wave vapour, and relative humidity (i.e., representing thermodynamics) for post-processing daily precipitation is in line with findings from previous studies showing that precipitation in this region is strongly associated with thermodynamical and dynamical interactions with topography (Orr et al., 2017; Potter et al., 2018; Bannister et al., 2019; Medina et al., 2010; Ziarani et al., 2019; Dimri et al., 2017). In contrast, somewhat surprisingly, input variables such as total and liquid precipitation have little influence on output values (Fig. 7), which we hypothesise is partly due to the redundancy that exists between input variables, which in turn enables GPR models to compensate for the ablation of a single input variable by leveraging other variables.

### 4.2 Downstream use of GPR post-postprocessed daily precipitation probability distributions

Probabilistic and deterministic predictions are inherently different and it is important to consider this when evaluating the quality of the products produced by both types of models. In this work, we are interested in assessing the performance of (probabilistic) GPR models, which leverage the pairwise correspondence between WRF daily precipitation output and observations, whilst capturing the predictive uncertainty that arises from multiple sources. A probabilistic MOS approach is justified if the goal is to fully leverage the richer predictions yielded by these models. For example, GPR model predictions can be used to map probabilities of exceedance for different precipitation thresholds (Fig. 8(a,b,c)), which form the basis of , early warning systems (Reichstein et al., 2025), infrastructure planning (Salem et al., 2020) and climate risk analysis (Jones and Mearns, 2005). This is particularly relevant for mountainous areas with steep terrain (e.g., HKH), where extreme precipitation drives hydrometeorological hazards such as floods and/or flash-floods, landslides, and avalanches (Haslinger et al., 2025; Dimri

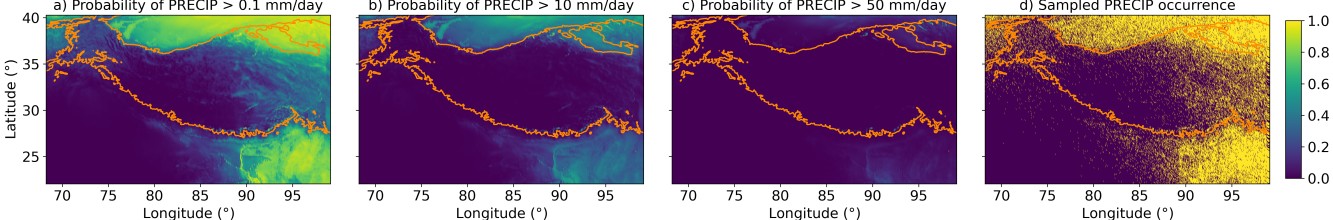

**Figure 8.** Panels a), b), and c) show maps showing the probability of precipitation exceeding 0.1 mm/day (a), 10 mm/day (b) and 50 mm/day (c) over the entire WRF spatial domain on an arbitrary date (10/10/2010) using the $MLP_L$ post-processing model trained using one of the k-fold splits in E2. Panel d) shows a map of modelled precipitation occurrence based on independently drawn samples from each grid-cell's predicted probability distribution for the entire WRF spatial domain on an arbitrary date (01/01/2010) using the $MLP_L$ post-processing model trained using one of the k-fold splits in E2. The 3000 m.a.s.l. contour is shown in orange in all four maps.

et al., 2017; Hunt and Dimri, 2021). More generally, probabilistic outputs provide a calibrated uncertainty estimate for RCM-modelled daily precipitation at each spatiotemporal location, offering useful information to impact modellers, especially given that raw RCM precipitation hindcasts do not exhibit direct pairwise correspondence with observations (Fig. B1). Additionally, there are downstream impact modelling settings that are capable of leveraging probability distributions over precipitation. The latter is an emerging but largely untapped area of research for many impact modelling fields reliant on precipitation as one of the main inputs (e.g., hydrological and crop modelling; Li et al., 2013; Peleg et al., 2017). Conversely, if the intended use of the models is to essentially reduce the probabilistic predictions into a deterministic product, then deterministic metrics should be used to evaluate them. To this end, we use ROC curves to show that, for different precipitation thresholds, deterministic projections of GPR predictions still consistently outperform WRF and $WRF_{SF}$ (Fig. 5).

Accurate representation of past and present-day daily precipitation holds significant importance for various downstream tasks, such as hydrological modelling, crop modelling, or hazard analysis, all of which heavily rely on the hindcast precipitation for calibration. In hydrological modelling, knowing the distribution and timing of daily precipitation is crucial for simulating streamflow and predicting river flooding events (e.g., Andermann et al., 2011; Huang et al., 2019; Li et al., 2017; Wulf et al., 2016). Similarly, in crop modelling, precise knowledge of daily precipitation patterns enables accurate estimation of water availability and crop growth, leading to improved yield predictions and agricultural management decisions (Wit et al., 2005). In hazard analysis, past climate data is paired with historic events (e.g., glacial lake outburst floods; Shrestha et al., 2023) to be able to determine what precipitation intensity triggered them. However, most impact modelling frameworks rely on the availability of spatio-temporal coherent precipitation fields, which the GPR framework does not directly support. This is a limitation of this study, and we discuss ways to address this in more detail below.

## 4.3 Limitations and future work

Local-scale variability of precipitation in HKH remains a challenge for MOS post-processing models aimed at ungauged locations. The diverse climatic conditions and complex terrain in the region result in contrasting precipitation patterns among

nearby stations, which limits the degree to which patterns observed in one location are representative of nearby locations (Immerzeel et al., 2014; Orr et al., 2017; Bhardwaj et al., 2021; Ren et al., 2017; Dimri et al., 2017; Thayyen et al., 2013; Potter et al., 2018). Using different locations to train, validate and test models is a common strategy in spatial prediction settings but it can introduce model biases (Burt et al., 2024), especially when the distribution of the gauging stations is uneven. Furthermore, the reliability of daily precipitation measurements in mountainous areas is compromised by issues such as gauge undercatch (Pritchard, 2021). Such challenges hinder our ability to thoroughly test MOS models across an entire region and thus to gain confidence in their use for operational post-processing of RCM daily precipitation.

Another important limitation of the study relates to the lack of spatial coherence when sampling from the predicted probability distributions. The GPR post-processing approach, as implemented in this work, assumes that the daily precipitation probability for each grid cell, conditional on a set of input features, is independent of its neighbours. This assumption, coupled with the large variability of precipitation in the region, leads to scattered precipitation occurrence maps when drawing independent samples from the predicted probability distribution at each location across the grid (Fig. 8d). There are several methods to introduce spatial and temporal coherence into probabilistic precipitation fields, including reordering techniques such as the Schaake Shuffle (Clark et al., 2004) as well as more advanced machine learning techniques such as latent variable models (Garnelo et al., 2018) and diffusion-based approaches (Yang et al., 2024; Turner et al., 2024). However, implementing these methods requires a reference dataset that captures realistic spatiotemporal correlations. In the context of HKH, given the sparse observational network, direct estimation of these dependencies from in-situ observations is not feasible. Instead, all such methods must rely on a pseudo-observational dataset to reconstruct coherent structures. We argue that the raw WRF precipitation outputs, which capture spatiotemporal precipitation structures, provide a natural choice for this role. In the case of the Schaake Shuffle, the weak conditioning of the probabilistic predictions on raw WRF precipitation – this is evidenced by the low influence WRF precipitation has on model outputs (Fig. 7, which can in turn be explained by the low pairwise correspondence between observed and WRF-simulated daily precipitation (Fig. B1) – supports using the RCM hindcast as the reference dataset for structured reordering. This method can be directly applied to a set of sampled values from the GPR output distributions at each location. Alternatively, i) the GPR framework can be extended by conditioning the post-processed daily precipitation distributions on a latent variable defined across space (and/or any other dimension, e.g., elevation) to build correlations between neighbouring locations, e.g., using Gaussian processes (Rassmussen and Williams, 2006), or ii) a diffusion model could be used to recover spatiotemporal coherent maps conditional on GPR-sampled precipitation fields. However, these approaches would also likely need to rely on the raw WRF precipitation fields for pre-training and thus capturing the spatio-temporal correlations.

Accurate representation of extreme precipitation events is another challenging area. Machine learning and statistical methods inherently perform best where there is sufficient training data; however, by definition, extreme precipitation events are relatively data-sparse. This is a well-known limitation of data-driven approaches and results in increased uncertainty in predictions at high precipitation thresholds compared to low thresholds. This is particularly true in the HKH region, where there is a pressing need for more station-based datasets, which would in turn increase the amount of observations for extreme events. Future work

could further explore the performance and added value of GPR post-processing for extreme precipitation events by focussing on a specific observed high-intensity precipitation event in the region.

Disaggregating results spatially and temporally is important to assess the extent to which different MOS post-processing models improve results at finer scales. In this work, we have presented results (dis)aggregated at the regional level, enabling an analysis of regional differences. However, further spatial granularity (e.g., at the station level) would potentially lead to a better

understanding of model performance across different elevations and latitudes. Seasonality of precipitation also varies greatly across the three study regions. For example, the winter westerlies are responsible for much of the annual precipitation in West UIB, whilst East UIB and Central UGB are summer monsoon dominated (Bookhagen and Burbank, 2010; Palazzi et al., 2013; Dimri et al., 2015). Given their relevance when it comes to impacts on water resources, flooding, and other rainfall-induced natural disasters, correctly post-processing dominating seasons for each region is another important aspect to assess. To some

extent, optimising GPR models to perform best across all seasons inherently weights seasons by their relative significance. However, this work does not explicitly optimise models by season. This is therefore a direction that could be explored in future work.

Although we considered various parametric distributions for modelling the conditional probability over daily precipitation and found that the Bernoulli gamma mixture model is a previously-used, robust, and effective choice (Williams, 1998; Cannon,

2008), we suggest further work be focused on the study of distributions for modelling the probability over daily precipitation (conditional on a set of variables). Finally, as we have shown that RCM post-processing patterns learned from one region may be relevant for post-processing other (poorly gauged or completely ungauged) regions, future effort should be devoted to investigating the benefits of applying transfer learning (Pan and Yang, 2010) and meta-learning (Vanschoren, 2018) techniques for mountainous / data sparse regions like HKH, i.e., involving a model being pre-trained using data from a set of regions

(especially those that are relatively data-rich, such as the Alps) and then fine-tuned for a different region or set of regions.

## 5  Conclusions

The compound effect of the local-scale variability and sparse observations that characterise daily precipitation in HKH poses a significant challenge when it comes to post-processing RCM outputs for ungauged locations. In this work, we address this issue by introducing a Generalised Probabilistic Regression (GPR) approach to MOS post-processing (i.e., downscaling and bias-

correcting) of RCM-simulated daily precipitation hindcasts for ungauged mountain locations using sparse in situ observations. We test the GPR approach across three HKH regions and four experiments that mimic real-world scenarios. These experiments explore the ability of GPR models to generalise to: 1) ungauged locations within each region using separate-region models, 2) ungauged locations within each region using combined-region models, 3) an entirely ungauged region using combined-region models, and 4) high-elevation ungauged locations within each region using separate-region models.

Overall, the three GPR model architectures we employ exhibit similar and consistently large performance improvements relative to both the WRF baseline and the WRF$_{SF}$ deterministic bias correction approach. Generalised linear models (VGLMs) are found to be a robust choice for GPR-based post-processing of WRF daily precipitation but non-linear models (MLP$_S$ and

MLP$_L$) do, in most cases, lead to improved performance. We find that GPR models are able to learn from sparsely distributed (both spatially and temporally) in situ observations and to generalise to new locations, using both separate-region or combined-region training settings. Performance of separate-region and combined-region GPR models is largely similar, resulting in much improved skill relative to WRF and WRF$_{SF}$. Combined-region GPR models are also capable of generalising to new (completely ungauged) regions by leveraging data from other two regions. Although there is an expected drop in performance compared to other experiments, this experiment still results in large skill improvements relative to WRF and WRF$_{SF}$, with simpler model architectures being more robust choices in this setting. Furthermore, we explore the degree to which GPR models are effective at post-processing ungauged high-elevation locations and find that their performance is consistent with previous experiments, suggesting that this approach could be used to better understand much of the ungauged high-elevation regions of HKH.

GPR model performance exhibits large regional variability, driven by a combination of factors including the availability of in situ observations, performance/bias of WRF baseline and hydro-meteorological characteristics of each region. Simple GPR models are best for large precipitation events. The differential influence that input variables have for different GPR models underscores the complex interplay between input features and GPR model architecture, with thermodynamic/cloud related input variables being especially important. Lastly, our results show that GPR models can use patterns learned from one region to improve RCM post-processing in other region, and we therefore suggest/hypothesise that transfer learning and meta-learning may be promising approaches to leverage observations/knowledge from data-rich mountain regions (e.g., Alps) to improve RCM post-processing in other (data-poor) regions.

*Code and data availability.* The code used to reproduce the experiments, generate figures, and analyse the results presented in this study is available at: https://github.com/mgironamata/pddp-mountains. The WRF simulation output is available via Norris et al. (2019). The SRTM elevation data is available at: https://earthexplorer.usgs.gov/. In situ gauge datasets were collected and provided by the Bhakra Beas Management Board and the Indian Meteorological Department (East UIB), the Pakistan Meteorological Department and the Water and Power Development Authority (West UIB), and the International Centre for Integrated Mountain Development (Central UGB). The authors of this paper do not have the required permission to make the gauge datasets publicly available but suggest that any readers interested in obtaining them contact the above organisations.

## Appendix A: Spike-and-slab mixture models

To explicitly model the probability over both precipitation occurrence (zero and non-zero values) and magnitude we resort to a special type of mixture models called spike-and-slab models. A spike-and-slab model for a random variable $Y$ is a generative model in which $Y$ attains some fixed value $v$ (spike) or is drawn from a probability distribution $p$ (slab).

To implement a spike-and-slab model, let us consider a set of inputs $x_{1:M}$ and outputs $y_{1:M}$. Now, let $r_{1:M}$ be a collection of M additional binary values, the $m$-th of which is 1 if $y_m > 0$ and 0 if $y_m = 0$. Assume that observations $y_{1:M}$ are drawn from a collection of random variables $Y_{1:M}$, respectively. Assume that $r_{1:M}$ are sampled independently from a Bernoulli distribution. Following that, $y_m$ is zero if $r_m$ is zero, and sampled from a continuous distribution with support $(0, \infty)$ (e.g., gamma or log-

normal) if $r_m$ is one. Below we detail the probabilistic models and associated conditional log-probability for various mixture models of this type.

If we choose a gamma distribution with parameters $\alpha_m$ and $\beta_m$ to model the continuous part of the distribution, $y_m$ is produced via the following probabilistic model:

$$r_m \sim \mathcal{B}(\pi_m),$$
$$(y_m|r_m = 1) \sim \Gamma(\alpha_m, \beta_m)$$

We call this model a Bernoulli gamma mixture model, for which the conditional log-probability of the collection of $(y,r)_{1:M}$ value pairs is given by:

$$\sum_{m=1}^{M} \log p(y_m, r_m|x_m)$$
$$= \sum_{m=1}^{M} \log \mathcal{B}(r_m; \pi_m) \Gamma(y_m; \alpha_m, \beta_m)^{r_m}$$
$$= \sum_{m=1}^{M} r_m[\log \pi_m + \log \Gamma(y_m; \alpha_m, \beta_m)] + (1-r_m)\log(1-\pi_m)$$

## Appendix B: Additional results

### B1 Daily precipitation thresholds

To justify the choice of precipitation intensity thresholds and illustrate the rarity of extreme precipitation events in the station data used in this study, Table B1 shows the percentage (and total number) of in situ observations exceeding specific daily precipitation thresholds, highlighting that: i) events exceeding 10 mm/day account for 2.56% of the data in West UIB, 9.26% in East UIB, and 6.17% in Central UGB; ii) at higher thresholds (30 and 50 mm/day), observations are limited — especially for West UIB and Central UGB, where less than 1% of observed events are above these thresholds; and iii) events exceeding 100 mm/day are extremely rare in all three regions, with the total number of observations being too low to justify the use of this threshold.

### B2 Additional evaluation metrics

The mean squared error (MSE) quantifies the squared difference between a set of predicted and observed value pairs. We calculate the MSE between the deterministic predictions $y_m$ and the observations $y_m^{obs}$. For probabilistic predictions, we use the mean of the predicted probability distribution as the predicted value $y_m$.

**Table B1.** Percentage (and total number) of in situ observations that exceed, or are equal to, a threshold of daily precipitation, for various thresholds (mm/day).

| Threshold (mm/day) | West UIB | East UIB | Central UGB |
|---|---|---|---|
| 0 (all events) | 100% (76,860) | 100% (364,713) | 100% (15,152) |
| 1 | 26.00% (19,983) | 20.08% (73,231) | 28.30% (4,288) |
| 10 | 2.56% (1,969) | 9.26% (33,779) | 6.17% (935) |
| 30 | 0.33% (252) | 2.95% (10,761) | 0.68% (103) |
| 50 | 0.08% (63) | 1.24% (4,511) | 0.26% (39) |
| 100 | 0.00% (1) | 0.22% (788) | 0.10% (15) |

$$\mathrm{MSE} = \frac{1}{M} \sum_{m=1}^{M} (y_m - y_m^{obs})^2 \tag{B1}$$

Using this, the MSE skill score (MSESS) is calculated as:

$$\mathrm{MSESS} = 1 - \frac{\mathrm{MSE}}{\mathrm{MSE_{WRF}}} \tag{B2}$$

The mean absolute error (MAE) quantifies the absolute difference between a set of predicted and observed value pairs. For deterministic predictions, we calculate the MAE between the predicted values $y_m$ and the observations $y_m^{obs}$. For probabilistic predictions, we use the mean values of the predicted probability distributions as predicted values $y_m$.

$$\mathrm{MAE} = \frac{1}{M} \sum_{m=1}^{M} |y_m - y_m^{obs}| \tag{B3}$$

Using this, we calculate the MAE skill score (MAESS) as:

$$\mathrm{MAESS} = 1 - \frac{\mathrm{MAE}}{\mathrm{MAE_{WRF}}} \tag{B4}$$

The threshold-weighted CRPS (twCRPS) (Gneiting and Ranjan, 2011, twCRPS) extends the CRPS by incorporating weight functions that emphasise specific portions of the predictive distribution. This modification allows for a tailored assessment of hindcast performance, particularly focusing on designated ranges within the support; in our case, extreme events with daily
precipitation above a threshold value $T$.

$$\mathrm{twCRPS} = \frac{1}{M} \sum_{m=1}^{M} u(y_m^{obs}) \int_{-\infty}^{\infty} (F(y_m) - H(y_m - y_m^{obs}))^2 \, dy_m. \tag{B5}$$

where:

$$u(y_m^{obs}) = \begin{cases} 0 & \text{if } y_m^{obs} < T \\ 1 & \text{if } y_m^{obs} \geq T \end{cases} \tag{B6}$$

Using this, the twCRPSS is then calculated as:

$$\text{twCRPSS} = 1 - \frac{\text{twCRPS}}{\text{twCRPS}_{\text{WRF}}} \tag{B7}$$

**Table B2.** MSE Skill Score (MSESS) values of post-processed daily WRF precipitation for the three GPR model architectures (VGLM, $\text{MLP}_S$ and $\text{MLP}_L$) and $\text{WRF}_{SF}$, calculated for all three regions and all four experiments. Higher MSESS values indicate better performance, with the best-performing MOS method for each experiment and region shown in bold. Note that $\text{MLP}_L$ for E3 West UIB was trained using a learning rate of $10^{-4}$ to ensure training convergence, while the other experiments used $10^{-3}$.

| Experiment | West UIB | | | | East UIB | | | | Central UGB | | | |
|---|---|---|---|---|---|---|---|---|---|---|---|---|
| | VGLM | $\text{MLP}_S$ | $\text{MLP}_L$ | $\text{WRF}_{SF}$ | VGLM | $\text{MLP}_S$ | $\text{MLP}_L$ | $\text{WRF}_{SF}$ | VGLM | $\text{MLP}_S$ | $\text{MLP}_L$ | $\text{WRF}_{SF}$ |
| E1 | 0.646 | **0.659** | 0.646 | 0.587 | 0.318 | **0.336** | 0.330 | -0.161 | 0.795 | **0.806** | 0.793 | 0.791 |
| E2 | 0.640 | **0.666** | 0.630 | 0.037 | 0.297 | **0.322** | 0.313 | 0.041 | 0.801 | 0.792 | **0.802** | 0.077 |
| E3 | 0.111 | **0.513** | 0.400 | -0.056 | **0.361** | 0.356 | 0.322 | 0.302 | 0.777 | 0.742 | **0.791** | -0.073 |
| E4 | 0.613 | **0.660** | 0.636 | 0.586 | 0.309 | **0.323** | 0.246 | -0.153 | 0.798 | **0.803** | 0.802 | 0.790 |

**Table B3.** MAE Skill Score (MAESS) values of post-processed daily WRF precipitation for the three GPR model architectures (VGLM, $\text{MLP}_S$ and $\text{MLP}_L$) and $\text{WRF}_{SF}$, calculated for all three regions and all four experiments. Higher MAESS values indicate better performance, with the best-performing MOS method for each experiment and region shown in bold. Note that $\text{MLP}_L$ for E3 West UIB was trained using a learning rate of $10^{-4}$ to ensure training convergence, while the other experiments used $10^{-3}$.

| Experiment | West UIB | | | | East UIB | | | | Central UGB | | | |
|---|---|---|---|---|---|---|---|---|---|---|---|---|
| | VGLM | $\text{MLP}_S$ | $\text{MLP}_L$ | $\text{WRF}_{SF}$ | VGLM | $\text{MLP}_S$ | $\text{MLP}_L$ | $\text{WRF}_{SF}$ | VGLM | $\text{MLP}_S$ | $\text{MLP}_L$ | $\text{WRF}_{SF}$ |
| E1 | 0.472 | 0.477 | **0.501** | 0.397 | 0.089 | 0.110 | **0.114** | -0.075 | 0.725 | 0.729 | **0.731** | 0.691 |
| E2 | 0.443 | **0.486** | 0.457 | 0.021 | 0.075 | **0.095** | 0.080 | 0.022 | 0.707 | 0.706 | **0.746** | 0.043 |
| E3 | 0.133 | 0.532 | **0.543** | -0.027 | 0.181 | **0.233** | 0.162 | **0.233** | 0.716 | 0.697 | **0.750** | -0.039 |
| E4 | 0.422 | **0.487** | 0.475 | 0.398 | 0.063 | **0.079** | -0.006 | -0.073 | **0.750** | 0.739 | 0.740 | 0.690 |

**Table B4.** Threshold-Weighted CRPS Skill Score (twCRPSS) values at 10 mm/day threshold for post-processed daily WRF precipitation for the three GPR model architectures (VGLM, $MLP_S$ and $MLP_L$) and $WRF_{SF}$, calculated for all three regions and all four experiments. Higher twCRPSS values indicate better performance, with the best-performing MOS method for each experiment and region shown in bold.

| Experiment | West UIB | | | | East UIB | | | | Central UGB | | | |
|---|---|---|---|---|---|---|---|---|---|---|---|---|
| | VGLM | $MLP_S$ | $MLP_L$ | $WRF_{SF}$ | VGLM | $MLP_S$ | $MLP_L$ | $WRF_{SF}$ | VGLM | $MLP_S$ | $MLP_L$ | $WRF_{SF}$ |
| E1 | 0.010 | **0.067** | 0.045 | -0.058 | 0.221 | 0.245 | **0.260** | -0.018 | 0.239 | **0.322** | 0.315 | 0.159 |
| E2 | 0.009 | 0.042 | **0.056** | 0.001 | 0.225 | 0.243 | **0.272** | 0.003 | 0.228 | 0.244 | **0.259** | 0.036 |
| E3 | **0.184** | -0.193 | -0.187 | -0.002 | **0.134** | 0.100 | 0.109 | -0.019 | **0.278** | 0.257 | 0.023 | -0.033 |
| E4 | -0.010 | 0.035 | **0.044** | -0.063 | 0.231 | 0.250 | **0.257** | -0.018 | 0.122 | 0.222 | **0.261** | 0.157 |

**Table B5.** Threshold-Weighted CRPS Skill Score (twCRPSS) values at 30 mm/day threshold for post-processed daily WRF precipitation for the three GPR model architectures (VGLM, $MLP_S$ and $MLP_L$) and $WRF_{SF}$, calculated for all three regions and all four experiments. Higher twCRPSS values indicate better performance, with the best-performing MOS method for each experiment and region shown in bold.

| Experiment | West UIB | | | | East UIB | | | | Central UGB | | | |
|---|---|---|---|---|---|---|---|---|---|---|---|---|
| | VGLM | $MLP_S$ | $MLP_L$ | $WRF_{SF}$ | VGLM | $MLP_S$ | $MLP_L$ | $WRF_{SF}$ | VGLM | $MLP_S$ | $MLP_L$ | $WRF_{SF}$ |
| E1 | -0.083 | **-0.002** | -0.018 | -0.109 | 0.126 | 0.147 | **0.166** | 0.006 | -0.395 | **-0.275** | -0.327 | -0.421 |
| E2 | -0.082 | -0.023 | **0.007** | -0.006 | 0.137 | 0.146 | **0.183** | -0.004 | -0.393 | -0.386 | -0.396 | **-0.007** |
| E3 | **0.093** | -0.163 | -0.139 | 0.004 | **0.009** | -0.023 | 0.003 | -0.095 | **0.278** | 0.257 | -0.539 | 0.001 |
| E4 | -0.092 | -0.031 | **-0.008** | -0.113 | 0.136 | 0.148 | **0.167** | 0.006 | -0.468 | -0.382 | **-0.333** | -0.423 |

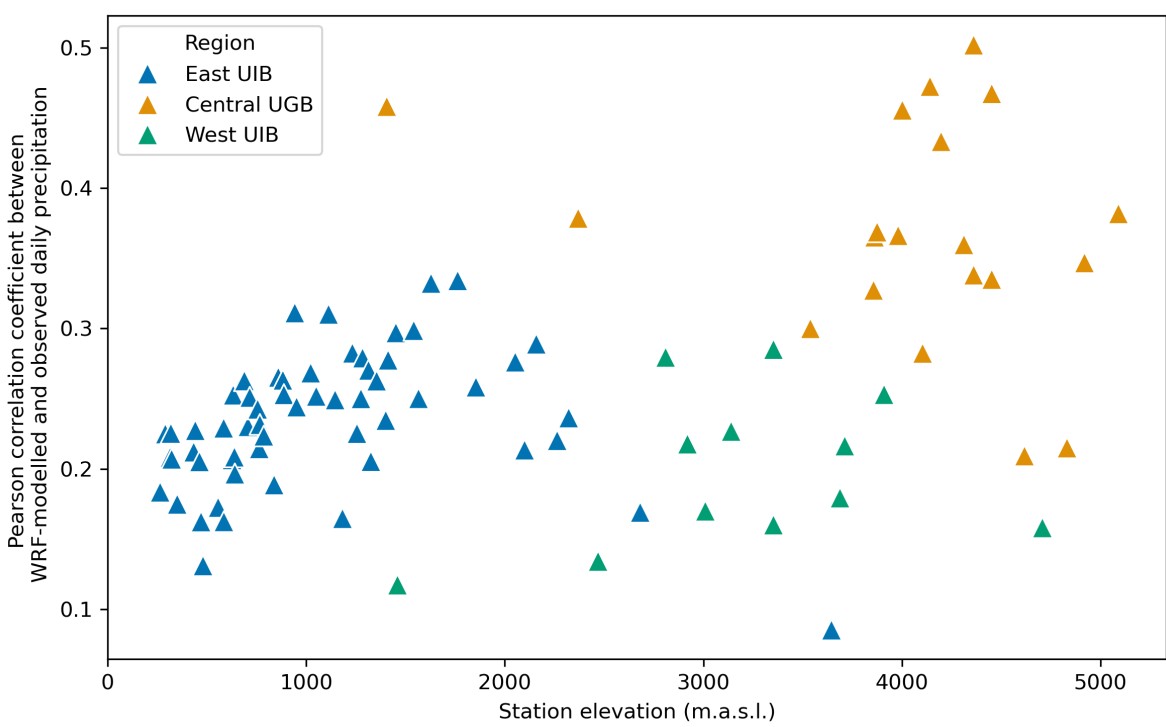

**Figure B1.** Pearson correlation coefficient between raw WRF-simulated and observed daily precipitation for each station, sorted by station elevation and coloured by region.

## Appendix C: Information Criteria for Model Selection

In order to assess the trade-off between the goodness-of-fit and model complexity, we consider various information criteria. In particular, we compute the Akaike Information Criterion (AIC; Akaike, 1973), as well as its small-sample corrected version (AICc; Sugiura, 1978) and the large-sample Kullback Information Criterion (KIC; Cavanaugh, 1999). These criteria are computed based on the maximum likelihood value and the number of parameters in the model. While our primary model evaluation relies on cross-validation and proper scoring rules, these criteria offer a complementary perspective, as discussed below.

The Akaike Information Criterion (AIC) is derived from an approximation of the Kullback–Leibler divergence between the true data generating process and a candidate model. It is given by

$$\text{AIC} = -2 \log L(\hat{\phi}) + 2k, \tag{C1}$$

where $L(\hat{\phi})$ denotes the maximum likelihood of the model (with parameters $\phi$), and $k$ is the number of free parameters. A lower AIC value indicates a model that is closer, in the Kullback–Leibler sense, to the true model. Moreover, when the sample size $n$ is small relative to the number of parameters $k$, the AIC tends to be biased. Therefore, a corrected version (AICc) is given by

$$\text{AICc} = \text{AIC} + \frac{2k(k+1)}{n-k-1}, \tag{C2}$$

is recommended in such cases. The AICc penalises models more strongly when $n$ is only moderately larger than $k$. In the limit as $n \to \infty$, AICc converges to AIC.

The Kullback Information Criterion (KIC) is based on the symmetric version of the Kullback's divergence and is defined as

$$\text{KIC} = -2 \log L(\hat{\phi}) + 3k. \tag{C3}$$

By using a penalty term of $3k$ rather than $2k$, the KIC generally imposes a more severe penalty for model complexity, thereby favoring models with fewer parameters.

For each candidate model, we compute total log-likelihood $\log L(\hat{\phi})$ by summing the per-observation contributions over the test set. In particular, we use the average per-observation negative log-likelihood ($\overline{\text{NLL}}$) values in Table 3 and the number of test samples $n$ in Table C1, to compute

$$\log L(\hat{\phi}) = -\overline{\text{NLL}} \times n. \tag{C4}$$

The information criteria are then computed via the above expressions, with $k$ values specified in Table C2.

**Table C1.** Test set sizes $n$ for each experiment and region.

| Experiment | West UIB | East UIB | Central UGB |
|---|---|---|---|
| E1, E2, E3 | 76,860 | 364,713 | 15,152 |
| E4 | 7,189 | 26,847 | 1,787 |

**Table C2.** Parameter counts $k$ for each candidate model.

| GPR model | Parameter count ($k$) |
|---|---|
| VGLM | 81 |
| $MLP_S$ | 303 |
| $MLP_L$ | 4,053 |

**Table C3.** AIC values for the three GPR model architectures (VGLM, $MLP_S$ and $MLP_L$), calculated for all four experiments and all three target regions. Lower AIC values indicate better quality, with the best GPR model for each experiment and region shown in bold.

| Experiment | West UIB | | | East UIB | | | Central UGB | | |
|---|---|---|---|---|---|---|---|---|---|
| | VGLM | $MLP_S$ | $MLP_L$ | VGLM | $MLP_S$ | $MLP_L$ | VGLM | $MLP_S$ | $MLP_L$ |
| E1 | 194310 | **190450** | 196567 | 860155 | **846011** | 846946 | 45012 | **44153** | 51138 |
| E2 | 194925 | **191987** | 198719 | 861614 | 843822 | **842569** | 45648 | **45365** | 51986 |
| E3 | **231049** | 436402 | 385796 | 1760996 | 1117357 | **1089845** | **51164** | 60759 | 67199 |
| E4 | 18609 | **18578** | 26093 | 63897 | **62891** | 71089 | 6863 | **6156** | 13506 |

**Table C4.** AICc values for the three GPR model architectures (VGLM, $MLP_S$ and $MLP_L$), calculated for all four experiments and all three target regions. Lower AICc values indicate better quality, with the best GPR model for each experiment and region shown in bold. Note that AICc is not defined for $k > n$ (i.e., $MLP_L$ for E4).

| Experiment | West UIB | | | East UIB | | | Central UGB | | |
|---|---|---|---|---|---|---|---|---|---|
| | VGLM | $MLP_S$ | $MLP_L$ | VGLM | $MLP_S$ | $MLP_L$ | VGLM | $MLP_S$ | $MLP_L$ |
| E1 | 194310.53 | **190452.61** | 197018.08 | 860155.29 | **846011.24** | 847037.02 | 45012.80 | **44165.26** | 54098.73 |
| E2 | 194925.41 | **191989.81** | 199170.16 | 861614.14 | 843822.96 | **842660.46** | 45649.19 | **45377.42** | 54947.24 |
| E3 | **231049.61** | 436404.61 | 386247.40 | 1760996.40 | 1117357.71 | **1089935.87** | **51164.51** | 60771.85 | 70159.85 |
| E4 | 18610.84 | **18605.26** | 36575.09 | 63897.27 | **62897.98** | 72530.81 | 6871.04 | **6280.65** | inf |

**Table C5.** KIC values for the three GPR model architectures (VGLM, $MLP_S$ and $MLP_L$), calculated for all four experiments and all three target regions. Lower KIC values indicate better quality, with the best GPR model for each experiment and region shown in bold.

| Experiment | West UIB | | | East UIB | | | Central UGB | | |
|---|---|---|---|---|---|---|---|---|---|
| | VGLM | $MLP_S$ | $MLP_L$ | VGLM | $MLP_S$ | $MLP_L$ | VGLM | $MLP_S$ | $MLP_L$ |
| E1 | 194391 | **190753** | 200620 | 860236 | **846314** | 850999 | 45093 | **44456** | 55191 |
| E2 | 195006 | **192290** | 202772 | 861695 | **844125** | 846622 | 45729 | **45668** | 56039 |
| E3 | **231130** | 436705 | 389849 | 1761077 | 1117660 | **1093898** | **51245** | 61062 | 71252 |
| E4 | **18690** | 18882 | 30146 | 63978 | **63194** | 75142 | 6944 | **6459** | 17559 |

*Author contributions.* MGM and AO defined the scope of the study. MGM conceptualised the probabilistic modelling approach with advice from SH and RET. MGM designed the experiments and conducted the research in close collaboration with AO, MW and RET. MGM and AO wrote the manuscript, with additional input from MW, RET, DB, DO, JN and JS. AO provided expertise on climate modelling, MW on post-processing and evaluation, and RET on probabilistic modelling and machine learning. DO and JS contributed to ensuring relevance for downstream applications. JN provided the WRF outputs. TP pre-processed the WRF outputs. DB, JS, and GHD provided the observation datasets for East UIB, Central UGB, and West UIB, respectively.

*Competing interests.* No competing interests are present.

*Acknowledgements.* This research was supported by the UKRI Centre for Doctoral Training in Application of Artificial Intelligence to the study of Environmental Risks (AI4ER) (EP/S022961/1). AO was funded by the UKRI/NERC grant 'The Big Thaw: gauging the past, present and future of our mountain water resources' (NE/X005267/1). RET is supported by the EPSRC Probabilistic AI Hub (EP/Y028783/1).

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
