# Peer review of "Probabilistic precipitation downscaling for ungauged mountain sites: a pilot study for the Hindu Kush Himalaya"

_EGUsphere, 2024_

## Author Response (AR1)

Dear Editors and Reviewers,

**Author response re. egusphere-2024-2805 (submitted to HESS)**

We thank you for your time in assessing our manuscript. We are pleased that both reviewers find the paper important, innovative, and well-written. We are also very grateful for their comments (as well as the Editor's note) on how the manuscript may be improved and strengthened. In particular, in response to the comments the revised manuscript includes:

i) Better characterisation of the (lack of) synchrony or pairwise correspondence between RCM-simulated and observed daily precipitation timeseries (Figure B1),

ii) Information on the number of events captured for each precipitation threshold, highlighting the relative lack of observations at high thresholds (Table B1).

iii) Use of additional evaluation metrics to, first, assess the results using deterministic metrics (Tables B2 and B3) and, second, evaluate the tails of the results, i.e., extremes (Tables B4 and B5),

iv) Use of information criteria for better understanding the trade-off between goodness-of-fit and model complexity,

v) More detailed discussion on the downstream uses of the post-processed probabilistic daily precipitation outputs, and

vi) More detailed discussion on limitations of the study, including on data availability for high precipitation thresholds and that the probabilistic results are spatially and temporally unconnected, and solutions to these issues.

We detail the changes we have made in response to the reviewers' suggestions below. Our responses are shown in blue, with quoted manuscript text shown in *italics* and added text marked in ***bold.*** Line references (e.g., lines 584-590) correspond to the tracked changes revised manuscript.

Best wishes

Marc Girona-Mata

**Reviewer 1**

Review for " Probabilistic precipitation downscaling for ungauged mountain sites: a pilot study for the Hindu Kush Karakoram Himalaya" by Marc Girona-Mata et al. submitted to EGUsphere (MS No.: egusphere-2024-2805).

General comments:

The authors present an innovative Generalized Probabilistic Regression (GPR) approach to Model Output Statistics (MOS) post-processing, specifically tailored for ungauged mountainous regions in the Hindu Kush Himalaya (HKH). By utilizing generalized linear models and artificial neural networks, the study convincingly demonstrates that GPR significantly improves the accuracy of daily precipitation estimates compared to raw regional climate model outputs and deterministic bias correction methods. This innovative approach of estimating daily precipitation addresses

critical challenges posed by sparse precipitation data and the complex topography of the HKH region. This work contributes substantially to the scientific understanding of precipitation patterns in data-scarce mountain regions. The GPR framework not only provides a robust tool for estimating daily precipitation but also quantifies uncertainties for regions with limited data availability. This method has practical implications, offering decision-makers a valuable resource for improved water management and disaster preparedness. The overall quality of the paper is high, adding both theoretical and applied value to the field of hydro-climatology in the data sparse and poorly understood region of the HKH. While the paper is well-written, incorporating the suggested improvements below would further enhance its impact.

Specific comments:

- **[R1-C1]** How does the GPR perform under extreme precipitation?

While the GPR method improves precipitation estimation, its performance declines with increasing daily precipitation, as seen in Fig. 4 (30 mm/day). In the eastern HKH, especially during the monsoon season, daily precipitation often exceeds 30 mm/day. How would GPR perform under extreme precipitation conditions? The authors could explore a case study involving an observed extreme precipitation event and evaluate the GPR method specifically for such conditions.

We thank the reviewer for this important point regarding the performance of GPR for extreme events. Machine learning (ML) and statistical methods inherently perform best where there is sufficient training data, and by definition, extreme precipitation events are data-sparse. This results in increased uncertainty in predictions at high precipitation thresholds, which is a well-known limitation of data-driven approaches in climate and weather modelling. We have mentioned this point in the revised manuscript in Section 4.3 (Limitations and future work) by adding the following text:

[lines 584-590]: *Accurate representation of extreme precipitation events is another challenging area. Machine learning and statistical methods inherently perform best where there is sufficient training data; however, by definition, extreme precipitation events are relatively data-sparse. This is a well-known limitation of data-driven approaches and results in increased uncertainty in predictions at high precipitation thresholds compared to low thresholds. This is particularly true in the HKH region, where there is a pressing need for more station-based datasets, which would in turn increase the amount of observations for extreme events. Future work could further explore the performance and added value of GPR post-processing for extreme precipitation events by focussing on a specific observed high-intensity precipitation event in the region.*

Despite this limitation, we robustly evaluated the GPR performance for extreme precipitation events in the manuscript using Brier scores (Table 4), reliability diagrams (Figure 4), and ROC scores (Figure 5) at multiple precipitation intensity thresholds, including 10 mm/day, 30 mm/day and 50 mm/day. To justify the choice of precipitation intensity thresholds and illustrate the rarity of extreme precipitation events in the station data used in this study, Table B1 (see below) shows the percentage (and total number) of in situ observations exceeding specific daily precipitation

thresholds. This table highlights that: i) events exceeding 10 mm/day account for 2.56% of the data in West UIB, 9.26% in East UIB, and 6.17% in Central UGB; ii) at higher thresholds (30 and 50 mm/day), observations are limited — especially for West UIB and Central UGB, where less than 1% of observed events are above these thresholds; and iii) events exceeding 100 mm/day are extremely rare in all three regions, with the total number of observations being too low to justify the use of this threshold. This confirms that the 10, 30 and 50 mm/day thresholds used in the manuscript represent extreme events. We have included Table B1 in Appendix B of the revised manuscript, and have updated the manuscript text in Section 2.6 (Evaluation metrics) as follows:

[lines 239-245]: ***The BSS metric is used to assess the ability of different post-processing methods to capture various precipitation thresholds (0, 1, 10, 30 and 50 mm/day) that span the spectrum of precipitation events, ranging from no precipitation to very extreme events. The frequency and total number of events exceeding these thresholds are included in Table B1 (Appendix B) showing that 10, 30 and 50 mm/day represent extreme precipitation events for which nevertheless some amount of observations are available (e.g., 0.33%, 2.95%, and 0.68% of the total number of events exceed the 30 mm/day threshold at West UIB, East UIB, and Central UGB, respectively), as opposed to higher thresholds such as 100 mm/day, which have less than 0.2% of observations, i.e., too low to justify use of this threshold.***

GIven the above, we deem that the three threshold-based metrics (along with the various thresholds) used in the manuscript already provide a thorough evaluation of the performance of GPR models for extreme precipitation events. However, we agree with the reviewer that further exploration of individual extreme events could yield additional insights, particularly in cases where the GPR method may offer added value by capturing peaks that are otherwise underrepresented in the WRF outputs. While a detailed case study is beyond the current scope of this study, partly because the current manuscript is already relatively dense, we acknowledge this as a promising direction for future work, and have added the following text in Section 4.3 (Limitations and future work) of the manuscript:

[lines 588-590]: ***Future work could further explore the performance and added value of GPR post-processing for extreme precipitation events by focussing on a specific observed high-intensity precipitation event in the region.***

- **[R1-C2]** How this can be used for wider benefits and decision making ?

The discussion section would benefit from elaborating on how the improved method can assist decision-makers. While the authors suggest that this method aids water resource management, the specifics remain unclear. For instance, could the authors detail practical applications such as how this method can be used in improving water resources related decision making ?

We thank the reviewer for this suggestion and the need for a clearer / more detailed discussion on how the probabilistic outputs yielded by the GPR post-processing framework can be used for better decision-making in the region. Following the reviewer's suggestion, we have expanded the discussion in Section 4.1 (Performance of GPR method for post-processing daily precipitation) of the manuscript as follows:

[lines 525-537]: [...] *A probabilistic MOS approach is justified if the goal is to fully leverage the richer predictions yielded by these models. For example, GPR model predictions can be used to map probabilities of exceedance for different precipitation* **thresholds** *(Fig. 8(a,b,c)), which form the basis of* ***early warning systems (Reichstein et al., 2025), infrastructure planning (Salem et al., 2020) and climate risk analysis (Jones and Mearns, 2005). This is particularly relevant for mountainous areas with steep terrain (e.g., HKH), where extreme precipitation drives hydrometeorological hazards such as floods and/or flash-floods, landslides, and avalanches (Haslinger et al., 2025; Dimri et al., 2017; Hunt and Dimri, 2021). More generally, probabilistic outputs provide a calibrated uncertainty estimate for RCM-modelled daily precipitation at each spatiotemporal location, offering insights for downstream impact models, especially given that raw RCM precipitation hindcasts do not exhibit direct pairwise correspondence with observations. Additionally, there are*** *downstream impact modelling settings that are capable of leveraging probability distributions over precipitation. The latter is an emerging but largely untapped area of research for many impact modelling fields reliant on precipitation as one of the main inputs (e.g., hydrological and crop modelling; Li et al., 2013; Peleg et al., 2017).*

Later on, in the same section, we provide specific examples of the importance of accurate representation of daily precipitation for various downstream applications, but also point out that the lack of spatio-temporal coherence that characterises GPR-modelled sampled precipitation fields can hinder their practical use:

[lines 541-550]: *Accurate representation of past and present-day daily precipitation holds significant importance for various downstream tasks, such as hydrological modelling, crop modelling, or hazard analysis, all of which heavily rely on the hindcast precipitation for calibration. In hydrological modelling, knowing the distribution and timing of daily precipitation is crucial for simulating streamflow and predicting river flooding events (Andermann et al., 2011, Huang et al., 2019, Li et al., 2017, Wulf et al., 2016). Similarly, in crop modelling, precise knowledge of daily precipitation patterns enables accurate estimation of water availability and crop growth, leading to improved yield predictions and agricultural management decisions (DeWit et al., 2005). In hazard analysis, past climate data is paired with historic events (e.g., glacial lake outburst floods; Shrestha et al., 2023) to be able to determine what precipitation intensity triggered them.* ***However, most impact modelling frameworks rely on the availability of spatio-temporal coherent precipitation fields, which the GPR framework does not directly support. This is a limitation of this study, and we discuss ways to address this in more detail below.***

Last, and prompted by Reviewer 2, we discuss ways to overcome the above mentioned limitation in Section 4.2 of the manuscript (see Reviewer 2's third comment for more details.)

Technical Corrections:

Here are technical comments P refer to page number while L refer to line number.

- **[R1-C3]** P1L17: Include the names of the major rivers originating from the HKH region.

Thank you for this suggestion. We have added the names of those major rivers originating from the HKH that are most relevant for this study. The updated text reads as follows:

[line 17]: *[...] many major rivers in South Asia, **such as the Indus or the Ganges,** supplying water resources to a rich [...]*

- **[R1-C4]** P2L21: The term "Hindu Kush Himalaya" (HKH) is widely recognized. Clarify why the authors include "Karakoram" in the name while retaining the abbreviation HKH.

We agree with the reviewer's suggestion and have, as a result, updated the manuscript to use the term Hindu Kush Himalaya, abbreviated as HKH.

- **[R1-C5]** P2L2021 and Fig 1: Add a background map showing the HKH's location in the context of the globe or Asia. Include major rivers. This would enhance the paper's appeal to a broader audience.

We thank the reviewer for this suggestion regarding Figure 1 and the inclusion of a background map showing the study region (Hindu Kush Himalaya) in a global or Asian context with major rivers. We carefully considered the content and configuration of this figure during the development of our manuscript and went through multiple iterations before finalising Figure 1 in its current form. Our rationale for the current configuration is twofold. First, our primary goal was to emphasise the study region, clearly highlighting the three target regions (West Upper Indus Basin, East Upper Indus Basin, and Central Upper Ganges Basin) along with the distribution of gauge measurements within each of these regions. We believe that adding a broader continental or global map would potentially shift the emphasis away from the specific regional details that are most relevant to our study. Second, while we acknowledge the value of providing a broader geographical context, we found that adding an inset map (such as a global or Asian reference map) made the figure overly complex and cluttered, reducing the clarity of the key information we intend to convey. Instead, we choose to focus on topography and gauge locations, which are critical to our analysis. Given these considerations, we have opted to maintain Figure 1 in its current form to ensure that it remains clear and effectively communicates the key geographic aspects mentioned above.

- **[R1-C6]** P2L25: What is meant by 'this precipitation knowledge gap..' not clear

This concept ("precipitation knowledge gap") refers to the idea outlined in the preceding sentence, which reads:

[lines 23-25]: *Yet, despite the large human populations depending on these resources for power, industry, tourism, farming, and domestic consumption, the contributions of rain and snow (and its ensuing melt) to these river systems are still poorly studied and little understood.*

**Reviewer 2**

This paper addresses an important topic of correcting output of daily precipitation from a regional climate model for sparsely gauged mountainous regions. The paper is well written. While the method is based on the unrealistic assumption of conditional independence spatially and temporally, it is understandable why the authors have done so. It was designed to make the best use of the very limited gauge data. The experiments were well designed. I have three main comments:

- **[R2-C1]** On evaluation, while it is OK to use CRPSS to compare various probabilistic results, it is not useful for comparing the probabilistic results with the deterministic results (WRF_SF). Deterministic results tend to produce inflated CRPS and therefore should be compared with median or mean of probabilistic results.

We appreciate the reviewer's comment regarding the evaluation of probabilistic and deterministic results. The primary goal of this study is to produce probabilistic precipitation predictions and evaluate them accordingly. As such, our primary focus is on probabilistic score metrics, particularly the Continuous Ranked Probability Score (CRPS), which is well-suited for comparing probabilistic approaches. We acknowledge that deterministic methods can produce inflated CRPS values when compared to probabilistic predictions. However, reducing the probabilistic outputs to their mean or median values for direct comparison with deterministic methods fundamentally changes the nature of the assessment, as it effectively converts a probabilistic forecast into a deterministic one—eliminating the very uncertainty information that the probabilistic approach is designed to capture.

However, we have nonetheless computed Mean Squared Error (MSE; Table B2, see below) and Mean Absolute Error (MAE; Table B3, see below) skill scores using the mean values of the probabilistic outputs and compared them to the deterministic WRF_SF outputs. This comparison is analogous to the CRPSS analysis presented in the manuscript (Table 4). These results show that GPR-based methods still outperform both the raw WRF outputs and the deterministic WRF_SF approach across all regions and experiments, regardless of the evaluation metric used (CRPS, MSE, or MAE). For deterministic outputs, CRPS is mathematically equivalent to MAE. Therefore, the $WRF_{SF}$ CRPS values in Table 4 are identical to its MAE values in Table B3. Moreover, while the relative ranking of MOS approaches (including the three GPR methods and WRF_SF) varies slightly across different metrics, all evaluation methods confirm that GPR outperforms both the raw WRF outputs (reference dataset) and the deterministic WRF_SF MOS approach. The MSE and MAE skill score results are fully consistent with the text discussing CRPS results (Table 4) in the manuscript.

We have included this additional information in the revised manuscript by adding tables B2 and B3 in Appendix B. We have also subsequently added the following text to Section 2.6 (Evaluation metrics) and Section 3 (Results) of the manuscript, respectively:

[lines 247-250]: ***To complement this, we also use the mean squared error (MSE) and mean absolute error (MAE) to compare the performance of the post-processed precipitation distributions reduced to their mean values against WRFSF outputs, by again calculating their associated skill scores MSESS and MAESS (see Appendix B).***

[lines 364 - 368]: ***Additionally, Tables B2 and B3 (Appendix B) show MSESS and MAESS values, respectively, for all three GPR models as well as $WRF_{SF}$. The results yielded by these metrics are in line with those in Table 4 (CRPSS), showing that while the relative performance of GPR models varies depending on the assessment metric, the best-performing GPR models still outperform $WRF_{SF}$ for all experiments and regions (except for E3 in East UIB, where $MLP_S$ and $WRF_{SF}$ yield the same MAESS value).***

Finally, we would like to note that we choose not to emphasize absolute CRPSS values to avoid overinterpretation, except for WRF_SF, as its CRPS values are directly comparable to the reference dataset.

- **[R2-C2]** Also on evaluation, I suggest that the authors should evaluate the tails of the results (substantive or extreme events) by using twCRPS (threshold-weighted CRPS). It is important to know how good the results are on high precipitation.

We thank the reviewer for this suggestion. We evaluate extreme precipitation events using threshold-based metrics, including Brier scores (Table 4), reliability diagrams (Figure 4), and ROC scores (Figure 5), applied at five precipitation intensity thresholds: 0, 1, 10, 30, and 50 mm/day. These metrics provide insight into the model's ability to predict extreme events in a probabilistic framework. However, in addition and following the reviewer's suggestion, in the revised manuscript we have now computed the threshold-weighted CRPS (twCRPS) using a delta function on observed precipitation values to assign weights. This approach is equivalent to filtering out observed events below the specified threshold (thereby reducing the number of available observations). The twCRPS results for the 10 mm/day and 30 mm/day thresholds (Tables B4 and B5) confirm that GPR-based methods outperform deterministic approaches (WRF$_{SF}$), consistent with the results obtained from the other evaluation metrics. Given that the manuscript already includes three different evaluation metrics for extreme events, and to maintain clarity and focus, we have opted to include the twCRPS results (Tables B4 and B5) in Appendix B rather than the main text. We have however also added the following text to Section 2.6 (Evaluation metrics) and Section 3 (Results) of the revised manuscript, respectively:

[lines 245-247]: ***In addition, we use the threshold-weighted CRPS (twCRPS; with precipitation thresholds of 10 and 30 mm/day) to further assess the tails of the predictive distributions, by calculating the associated skill score twCRPSS (see Appendix B). The CRPSS, BSS and twCRPS metrics are also used to evaluate the skill of WRF$_{SF}$.***

[lines 385-391]: ***In addition, Tables B4 and B5 (Appendix B) include twCRPSS values using 10 and 30 mm/day thresholds, respectively, for the three GPR models and WRF$_{SF}$. These results, which provide further insight regarding the ability of post-processing methods to capture extreme precipitation events, show that MLP models are best at characterising the tails of the predictive distributions in E1, E2, and E4 (excluding E2 for Central UGB in Table B5, where WRF$_{SF}$ outperforms GPR models), whereas VGLM is the most robust option in E3. In West UIB and Central UGB, twCRPSS values for thresholds of 30 mm/day show all methods performing poorly; this can be explained by the relative lack of observations for events exceeding this threshold in both regions (Table B1).***

- **[R2-C3]** The probabilistic results are spatially and temporally unconnected, making it difficult for the users to apply the results. The authors should carefully consider this aspect and recommend how the results can be used when spatial and temporal connections are needed, as is the case in most applications.

We appreciate the reviewer's comment regarding the lack of spatial and temporal coherence in the probabilistic outputs, which is one of the most important limitations of the GPR MOS post-processing method. The GPR method assumes conditional independence, meaning that

probabilistic predictions at each grid cell are independent given the predictors (i.e., WRF outputs and other input variables, listed in Table 2). As the reviewer correctly points out, this assumption is a necessary consequence of the sparse observational dataset, which does not capture (and thus does not allow direct estimation of) inter-site dependencies. Consequently, when drawing independent samples from the probabilistic distributions at each location, no explicit spatial or temporal structure is imposed on the sampled precipitation fields.

Notwithstanding the above, the probabilistic outputs still retain implicit spatial and temporal consistency through the predictor variables. Neighbouring grid cells with similar predictor values will tend to have similar probability distributions, even though individual random draws from those distributions may lack coherence. As a result, the probabilistic outputs can be used to create exceedance probability maps for specific precipitation thresholds (e.g., 1 mm/day, 10 mm/day, or 50 mm/day; shown in Figure 8), which are critical for risk assessment, early warning systems, and infrastructure planning, particularly in regions where deterministic precipitation estimates are highly uncertain. Additionally, and more generally, probabilistic outputs provide a calibrated uncertainty estimate for RCM-modelled daily precipitation at each spatiotemporal location, offering insights for impact models—especially given that raw RCM precipitation hindcasts do not exhibit direct pairwise correspondence with observations (see below Figure B1, which we have also included in Appendix B in the revised manuscript).

Several methods exist to introduce spatial and temporal coherence into probabilistic precipitation fields, including reordering techniques (e.g., the Schaake Shuffle), latent variable models, and diffusion-based approaches. However, implementing these methods requires a reference dataset that captures realistic spatiotemporal correlations. Given the sparse observational network, direct estimation of these dependencies from in-situ observations is not feasible. Instead, all such methods must rely on a pseudo-observational dataset to reconstruct coherent structures. The raw RCM dataset, which inherently captures spatiotemporal precipitation structures, provides a natural choice for this role.

However, using the raw RCM precipitation outputs as a reference dataset for building spatio-temporal correlations into the predictions can be problematic given the predictions are already conditioned on RCM precipitation. Additionally, it is important to note that the predictions are only 'weakly' conditioned on raw RCM precipitation outputs, as shown by the low influence of RCM precipitation on predictions (Figures 6 and 7). This can in turn be explained by the low pairwise correspondence exhibited between RCM precipitation and observed precipitation (Figure B1), further supporting the idea that GPR probabilistic outputs capture uncertainty beyond just reflecting (deterministic) model biases.

In the case of the Schaake Shuffle, the weak conditioning of the probabilistic predictions on (raw) RCM precipitation supports using the raw RCM hindcast as the reference dataset for structured reordering. Alternatively, more advanced ML techniques, such as latent variable models or diffusion-based methods, could leverage the raw RCM precipitation fields for pre-training, allowing the learned spatial dependencies to be imposed onto the probabilistic outputs.

We have updated the discussion section of the manuscript to include this information, in particular Section 4.2 (previously 4.1), which has been re-titled to "Downstream use of GPR post-processed daily precipitation probability distributions". (Note, the structure of the discussion section has been updated and a new section "4.1 Performance of GPR method for post-processing daily precipitation" added as a result.) A detailed overview of the changes in Section 4.2 can be found in our response to Reviewer 1's second comment (R1-C2). In addition, we have also expanded Section 4.3 (Limitations and future work; previously 4.2) by adding the following text:

[lines 565-590]: ***There are several methods to introduce spatial and temporal coherence into probabilistic precipitation fields, including reordering techniques such as the Schaake Shuffle (Clark et al., 2004), as well as more advanced machine learning techniques such as latent variable models (Garnelo et al., 2018) and diffusion-based approaches (Yang et al., 2024; Turner et al., 2024). However, implementing these methods requires a reference dataset that captures realistic spatiotemporal correlations. In the context of HKH, given the sparse observational network, direct estimation of these dependencies from in-situ observations is not feasible. Instead, all such methods must rely on a pseudo-observational dataset to reconstruct coherent structures. We argue that the raw WRF precipitation outputs, which capture spatiotemporal precipitation structures, provide a natural choice for this role. In the case of the Schaake Shuffle, the weak conditioning of the probabilistic predictions on (raw) WRF precipitation supports using the RCM hindcast as the reference dataset for structured reordering. This method can be directly applied to a set of sampled values from the GPR output distributions at each location. Alternatively, i) the GPR framework can be extended by conditioning the post-processed daily precipitation distributions on a latent variable defined across space (and/or any other dimension, e.g., elevation) to build correlations between neighbouring locations, e.g., using Gaussian processes (Rassmussen and Williams, 2006), or ii) a diffusion model could be used to recover spatiotemporal coherent maps conditional on GPR-sampled precipitation fields. However, these approaches would also likely need to rely on the raw WRF precipitation fields for pre-training and thus capturing the spatio-temporal correlations.***

***Accurate representation of extreme precipitation events is another challenging area. Machine learning and statistical methods inherently perform best where there is sufficient training data; however, by definition, extreme precipitation events are relatively data-sparse. This is a well-known limitation of data-driven approaches and results in increased uncertainty in predictions at high precipitation thresholds compared to low thresholds. This is particularly true in the HKH region, where there is a pressing need for more station-based datasets, which would in turn increase the amount of observations for extreme events. Future work could further explore the performance and added value of GPR post-processing for extreme precipitation events by focussing on a specific observed high-intensity precipitation event in the region.***

Lastly, we have also added the following text in Section 3 (Results) explaining Figure B1:

[lines 312 - 316]: **In addition, Figure B1 (Appendix B) shows the Pearson correlation coefficient between raw WRF-simulated and observed daily precipitation timeseries for**

**the different stations, showing that while regional differences exist (e.g., correlations in Central UGB, where stations are relatively close to each other, are higher than in East UIB and West UIB, where the distance between is stations are larger) the pairwise correspondence (or synchrony) between WRF and observations is low, ranging between 0.1 and 0.5.**

**Editor's comment**

- I would encourage the Authors to consider adoption of formal model discrimination criteria to assess the skill (in a relative sense) of the models considered. It is of course my intention not to disregard any of the constructive comments emerged during this phase.

We thank the Editor for this suggestion to assess the GPR models' overall quality by using information criteria that consider both the models' capacity to represent the data as well as their complexity. Whilst our model evaluation approach relies on cross-validation and proper scoring rules, these criteria can indeed offer a complementary perspective. Consequently, we have computed the Akaike information criterion (AIC), its corrected version for small sample sizes (AICc), and the Kullback information criterion (KIC). These criteria consider both the log-likelihood of the model and the number of parameters in the model (Table C2 in Appendix C). The results of this analysis are included in Tables C3, C4 and C5 in Appendix C. Additionally, the revised manuscript includes text explaining the methods (in Section 3), outlining the results (in Section 4) and discussing the findings (in Section 5) of this analysis.

In particular, we have added the following text in Section 3 of the manuscript:

[revised manuscript text omitted]